# Sexually dimorphic renal expression of mouse *Klotho* is directed by a kidney-specific distal enhancer responsive to HNF1b

Jakub Jankowski [1,4] ✉, Hye Kyung Lee [1], Chengyu Liu[2], Julia Wilflingseder[3] & Lothar Hennighausen[1]

Transcription enhancers are genomic sequences regulating common and tissue-specific genes and their disruption can contribute to human disease development and progression. *Klotho*, a sexually dimorphic gene specifically expressed in kidney, is well-linked to kidney dysfunction and its deletion from the mouse genome leads to premature aging and death. However, the sexually dimorphic regulation of *Klotho* is not understood. Here, we characterize two candidate *Klotho* enhancers using H3K27ac epigenetic marks and transcription factor binding and investigate their functions, individually and combined, through CRISPR-Cas9 genome engineering. We discovered that only the distal (E1), but not the proximal (E2) candidate region constitutes a functional enhancer, with the double deletion not causing *Klotho* expression to further decrease. E1 activity is dependent on HNF1b transcription factor binding site within the enhancer. Further, E1 controls the sexual dimorphism of *Klotho* as evidenced by qPCR and RNA-seq. Despite the sharp reduction of *Klotho* mRNA, unlike germline *Klotho* knockouts, mutant mice present normal phenotype, including weight, lifespan, and serum biochemistry. Lastly, only males lacking E1 display more prominent acute, but not chronic kidney injury responses, indicating a remarkable range of potential adaptation to isolated *Klotho* loss, especially in female E1 knockouts, retaining renoprotection despite over 80% *Klotho* reduction.

Transcriptional enhancers allow for increased expression of genes under their control and help build tissue-specific expression profiles[1]. Enhancers bind transcription factors and facilitate the subsequent construction of the mediator complex forming a bridge between enhancer and the gene promoter. The advent of next-generation sequencing methods, such as ChIP-seq, has enabled the visualization of enhancer histone modifications, DNA binding proteins, including transcription factors, and RNA Pol II. There are multiple databases gathering enhancer mutations relevant to human disease and making them an appealing target of investigation[2,3]. However, due to the diverse structure, range, and orientation of enhancers relative to their target genes, experimental validation of potential enhancers is a labor-intensive yet essential aspect of unraveling the gene regulatory landscape.

Epigenetic regulation of renal gene expression is an active target of investigation. Cornerstone literature on acute kidney injury, polycystic kidney disease, and diabetes were published only recently and illustrated hundreds of candidate transcription elements changing activity and shaping the response to renal pathology[4–7]. Among the transcription factors responsible for those changes, HNF1b is inextricably linked to kidney health[8]. It plays a role in fetal development, ion transport, and mitochondrial health, while human mutations of HNF1b result in various nephropathies[9–12]. Taken together, those insights into renal enhancer activity open new avenues of investigation, as in some cases, the usual gene deletion approach is not viable to dissect the gene's role, and reducing gene expression through enhancer deletion can provide a new understanding of clinically relevant mechanisms.

The role of the *Klotho* gene, first described in 1997[13]. was initially explored in gene knockout mice, which displayed acute phenotypes resembling ageing. Accelerated osteoporosis, hyperphosphatemia, skin

[1]Section of Genetics and Physiology, Laboratory of Cellular and Molecular Biology, National Institute of Diabetes and Digestive and Kidney Diseases, US National Institutes of Health, Bethesda, MD, 20892, USA. [2]Transgenic Core, National Heart, Lung, and Blood Institute, US National Institutes of Health, Bethesda, MD, 20892, USA. [3]Department of Physiology and Pathophysiology, University of Veterinary Medicine Vienna, 1210 Vienna, Austria. [4]Present address: 8 Center Drive, Room 107, 20892 Bethesda, MD, USA. ✉e-mail: jakub.jankowski@nih.gov

atrophy, and infertility of both sexes indicated that *Klotho* was crucial for survival[14,15]. The combination of later studies dissecting *Klotho* transcripts and their impact on phosphate-calcium homeostasis[16,17], soon elevated it to the status of a potential kidney injury marker, though it is still debated whether KLOTHO's depletion is a cause or the consequence of disease development[18].

Despite the repeated observation that *Klotho* expression decreases in kidney injury setting[19], the exact mechanisms governing this change are unknown. Additionally, because 50% of *Klotho* knockout mice die at six weeks of age, investigating Klotho in adult research animals remained a challenging task. While renal proximal and distal tubule, as well as parathyroid-specific *Klotho* knockout mice exist[20–23], their phenotype is much milder, and it's unknown whether they properly recapitulate the intricacies of human disease. A mouse model where gene expression is evenly reduced but not null throughout the body is likely to be a more robust tool to investigate gene function and the degree of decreased expression's impact on phenotype, than tissue-restricted or inducible ones[24].

An additional challenge in the assessment of the function of *Klotho* is its sexually dimorphic expression. Female mice are underrepresented in the kidney injury literature, as their sex is linked with renoprotection against acute injury. A similar difference is observed between men and women, with the latter often indicated as less likely to suffer acute injury, but more prone to developing chronic disease[25,26]. It is recognized that male mice express more *Klotho* mRNA, and recent reports link high testosterone and Klotho levels, suggesting the relevance of experimental data for clinical settings and the need to better understand the regulatory mechanisms of *Klotho*[27,28].

In this manuscript, we follow upon our previous study, where we dissected the renal transcriptional landscape before and after injury using RNA-seq and ChIP-seq, identifying hundreds of potential gene regulatory elements[4]. We investigate the hypothesis that two DNA elements located upstream of the *Klotho* gene, displaying a decrease in activity caused by injury accompanied by lower gene expression, are functional enhancers of *Klotho*. We constructed several lines of mice carrying deletions of different sizes in the two candidate enhancers and investigated the biological consequences of the deletions on gene expression, sexual dimorphism, and the effects of enhancer depletion on acute kidney injury and fibrosis models. Mice lacking one of the enhancer candidates displayed a significant decrease in *Klotho* expression but had no premature aging phenotype and were fertile, unlike *Klotho* knockout mice. The effect of the deletion was dependent on impacting the HNF1b transcription factor binding site, illustrating its direct link to *Klotho* regulation. Female renal *Klotho mRNA levels* were impacted much stronger than male, further intensifying the inherent dimorphism in gene expression. Surprisingly, though Klotho deficiency resulted in higher susceptibility to male ischemia-reperfusion injury, the development of fibrosis was not impacted by *Klotho* levels, and female mice retained their relative resilience to kidney injury despite markedly lower *Klotho* levels.

## Results
### Candidate *Klotho* enhancers and construction of the enhancer-deficient mice
The two candidate *Klotho* enhancers are located approximately 40 kbp upstream of the gene body and promoter region (Fig. 1a). Those sequences are characterized by H3K27ac marks[4] and the occupation by transcription factors, including hepatocyte nuclear factor 1b (HNF1b), glucocorticoid receptor (GR) and the estrogen-related receptor gamma (Errγ), further marking them as potential enhancers (Fig. 1b). Only a known HNF1b motif[29] can be found within the enhancer sequences. Pol II coverage of the enhancer loci is observed as well, suggesting transcription and potential presence of enhancer RNAs (eRNAs), however little to no total RNA-seq reads map to the enhancer loci (Supplementary Fig. 1). The *Pds5b* gene, located further upstream, is expressed at a relatively lower level than *Klotho* (mean 1686 vs. 21134 WT male read counts as measured by RNA-seq, Supplementary Spreadsheet 1) in renal tissues and has no established impact on kidney disease[30], thus is unlikely to be regulated by the putative enhancers. In addition to the relative proximity of the enhancers to the gene and

transcription factor occupancy, height of the H3K27ac peaks were observed to lower after renal injury in tandem with the gene promoter marks, suggesting their role in gene expression modulation[4].

To distinguish between the two candidate enhancer regions, we called the distal one E1 and the proximal one E2. Initially, we attempted to excise the entire E1 fragment using two sgRNAs in the CRISPR-Cas9 system. However, two attempts at generating a mouse yielded no viable pups for undetermined reasons. We only obtained viable mice after targeting the HNF1b motif located within E1. Due to the imprecise nature of CRISPR-Cas9[31], we observed a range of deletions in the desired locus and developed four separate mouse lines, further called 1744 del (or E1 KO), 1001 del, 145 del, and 31 del. A wild-type (WT) mouse line was generated in parallel, by crossbreeding heterozygous E1 KO mice. Deletion of E2, targeted with two sgRNAs, resulted in a 900 bp deletion in the desired region (E2 KO). We chose the E1 KO mice as the ones best representing the entire E1 deletion and used them as a source to obtain mice lacking both E1 and E2 (E1/E2 KO), aiming to investigate any compensatory mechanisms a singular deletion might have. To confirm the effects of the deletion on the enhancer structure and activity, we performed ChIP-seq to assess acetylation and methylation of the enhancer and promoter regions (Fig. 1c). We observed that while E2 deletion retains E1 marks, E1 deletion visibly diminishes E2 peak as well (Fig. 1d).

### *Klotho* enhancer function is dependent on HNF1b binding
First, we investigated whether the enhancer deletions impacted *Klotho* expression and mRNA levels and whether the size of the deletion impacted their effects. The range of E1 deletions relative to the HNF1b binding site is shown in Fig. 2a. The E1 (1744 bp) and 145 bp deletions cover the entire HNF1b motif, while the 31 bp deletion only removes half of the binding motif. The 1001 bp deletion begins 8 nucleotides downstream of the binding site, leaving it intact. Only the mouse lines where the HNF1b binding site was disrupted displayed significantly lower (~50%) renal *Klotho* expression levels (Fig. 2b). The 1001 bp deletion, despite removing a significant portion of the putative enhancer, did not cause a decrease of *Klotho*. Surprisingly, complete deletion of the proximal E2 enhancer had no effect on gene expression (Fig. 2c), suggesting that this sequence is not an enhancer in its own right. To investigate whether E2 could function synergistically with E1, we generated mice lacking both E1 and E2. Notably, *Klotho* expression in E1/E2 KO mice was similar to E1 KO, suggesting that E2 has no significant impact on *Klotho* expression and elicits no compensatory effect in the absence of E1 (Fig. 2d).

Next, we visualized the mouse and human *Klotho* acetylation marks to investigate whether we can observe analogous peaks and confirm the role of HNF1b (Supplementary Fig. 2a). We did not observe corresponding peaks, however, two peaks relatively further upstream than E1 were present in human ChIP-seq data. We used FIMO tool[32] to investigate whether HNF1b motifs are present in those loci and found seven matches, indicating a potential for HNF1b activity (Supplementary Spreadsheet 2). Further, we aligned UCSC Genome Browser[33] to the same region and saw that both the mouse enhancers and human peaks are relatively conserved among rodents and primates (Supplementary Fig. 2b).

### Expression of *Klotho* in females lacking the E1 enhancer decreases by 90% without visible physiological consequences
Since *Klotho* gene expression is sexually dimorphic[34] it was imperative to investigate the physiological consequences of the enhancer deletions in both sexes. All the obtained mutant lines displayed no significant deviation in phenotype compared to wildtype. As opposed to *Klotho* knockout mice, they had no shortened lifespan, reaching over a year with no unusual health issues (only rare cases of dermatitis characteristic for the background at >6 months old were observed) and gained weight normally (Supplementary Fig. 3). Both WT and E1 KO female mice displayed decreased renal *Klotho* expression compared to males, approximately 50 and 90%, respectively (Fig. 3a). In combination with the baseline 50% decrease in renal *Klotho*, this means that in similar injury models, female mice might be more severely affected. Next, we found that serum FGF23, the main cofactor of Klotho[35],

**Fig. 1 | *Klotho* locus includes two putative enhancers.** Histone modification and Pol II ChIP-seq data of *Klotho* (red) locus on mouse chromosome 5 (**a**). Candidate enhancers are marked by H3K427ac peaks and differentiated from gene promoters through the lack of H3K4me3 peaks. Closeup of *Klotho* enhancers E1 and E2, as well as transcription factors binding within them (red), relative to *Klotho* promoter (gray) (**b**). Representative histone modification ChIP-seq showing the extent of the deletions (red) in male single enhancer and double knockout mice and a closeup (**c**, **d**).

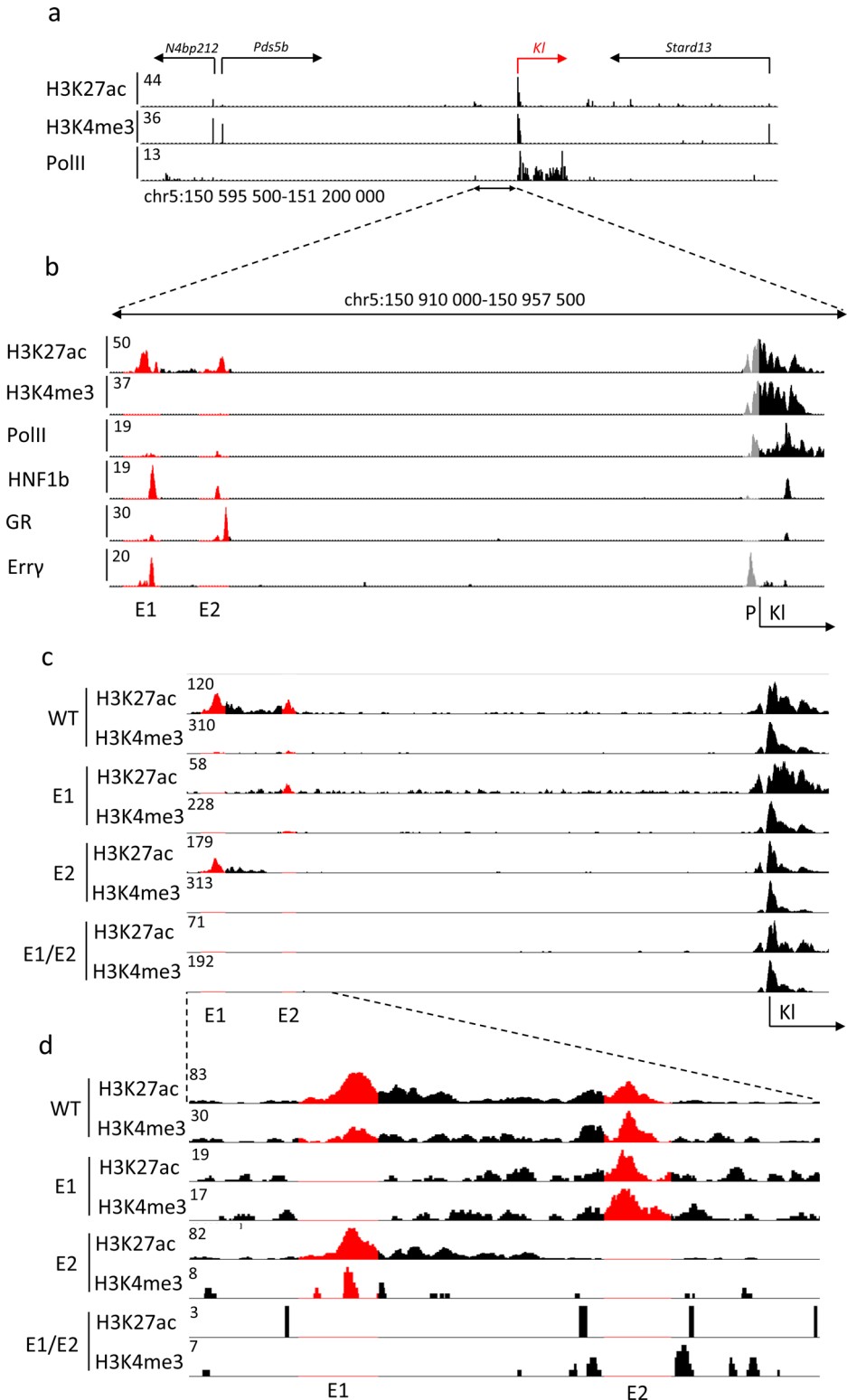

while remaining at the baseline level in male E1 KO mice, is significantly elevated in females (Fig. 3b). Assuming that this difference might be bound to androgen and estrogen activity, we measured *Klotho* expression in kidneys of pre-pubescent mice and found a disparity mirroring adults, though less pronounced in E1 KO mice, most likely due to lower overall expression levels (Supplementary Fig. 4a). Additionally, we performed ovariectomy in WT mice at 3 weeks old and euthanized them after 5 weeks to investigate potential impact of female sex hormones on *Klotho* expression but observed normal expression levels at 8 weeks old (Supplementary Fig. 4b).

Next, we assessed the activity at the enhancer and promoter elements of *Klotho* in the E1 deletion mice, as those changes can help narrow down the causes of sexually dimorphic gene expression. To visualize the *Klotho* enhancer and promoter regions, we performed ChIP-seq and saw that active H3K27ac and H34Kme3 promoter marks are being preserved in male E1 KO mice compared to females (Fig. 3c). We observed loss of H3K27ac marks in both E1 and E2 regions upon E1 deletion, indicating a link between E1 and E2, despite apparent irrelevance of E2 for *Klotho* expression regulation, and E2 deletion didn't lower the gene expression below baseline in

**Fig. 2 | Enhancer deletions impacting the HNF1b transcription factor binding site have a marked effect on *Klotho* expression.** Coverage of E1 deletions relative to HNF1b transcription factor binding site originally targeted by sgRNA (**a**). qPCR of renal *Klotho* expression levels in wild type and mutant E1 mouse lines with strains where HNF1b binding site is impacted in red (**b**). qPCR of renal *Klotho* expression levels in wild type and mutant E2 mouse line (**c**) and wild type, E1, and double knockout mouse lines (**d**). Expression is shown as fold change compared to control. Bar = SEM; **b, c** *n* = 4, d *n* = 6, One-way ANOVAs with group mean comparisons, ***P < 0.001, ****P < 0.0001.

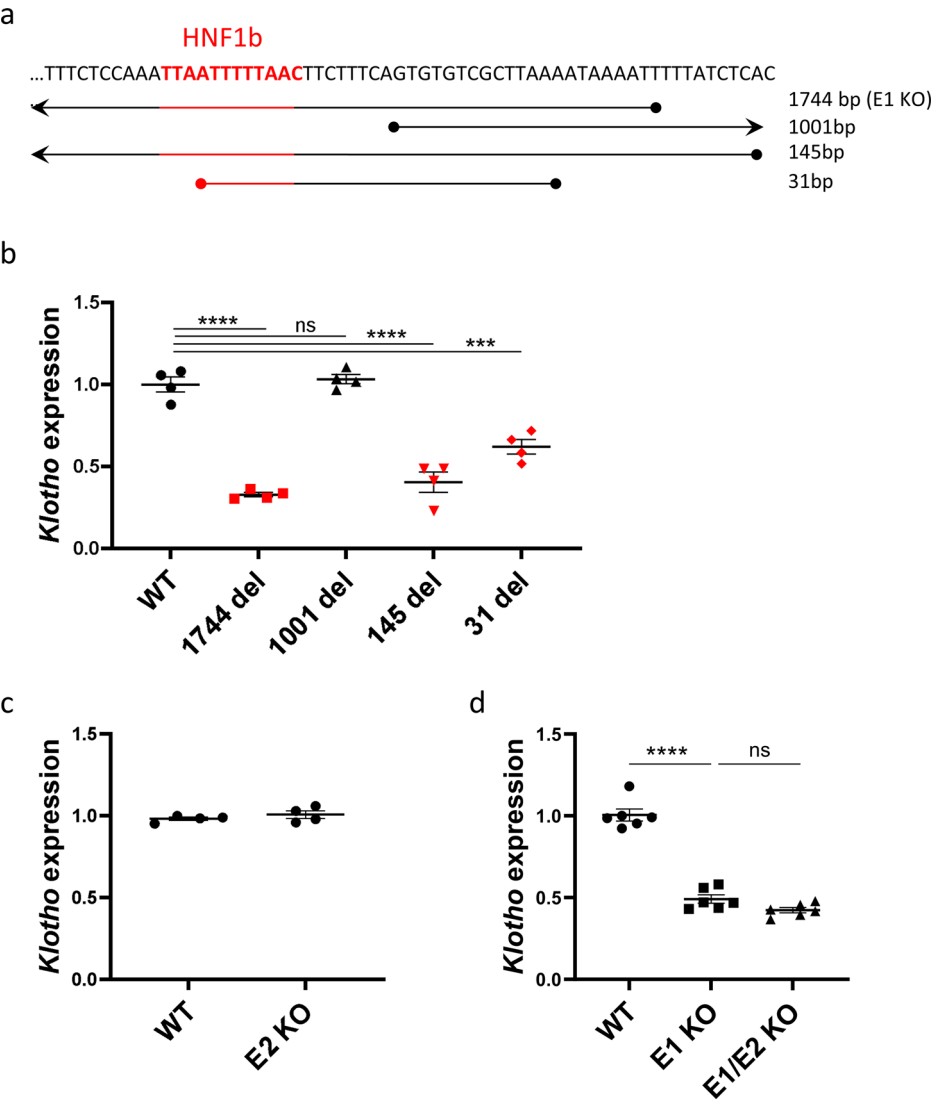

females as well (Supplementary Fig. 4c). Despite the increase in FGF23 and apparent sexual dimorphism, serum phosphate, and calcium, as well as other serum components (albumin, bicarbonate, blood urea nitrogen, chloride, creatinine, glucose, potassium, sodium, and total protein) in all experimental groups, remained at the same, physiological level (Fig. 3d, e and Supplementary Fig. 5).

**Transcriptional consequences of E1 deletion**

To investigate genetic programs altered by E1 deletion, we performed bulk RNA-seq on male and female WT and E1 KO kidneys with the goal of observing organ-wide changes. Even at the baseline, WT male and female kidneys are significantly sexually dimorphic, with 939 deregulated genes (DEGs) (Fig. 4a and Supplementary Spreadsheet 1). Significant difference in *Klotho* expression is clearly visible in E1 male vs. female comparison (mean read count 12656 vs. 4685 respectively), among a similar number of 1021 DEGs. The appearing difference strongly suggests that E1 deletion impacts females more. Within sexes, there are only 42 DEGs in male WT vs. E1 and 57 in females, with *Klotho* holding the highest significance score in the latter group, but not clearing the minimum twofold decrease threshold in males required to be included, thus again indicating strong sexual dimorphism. Small amount of DEGs and the fact that this last two, sex-dependent DEG lists only have three common elements (The *Gulo* gene and two non-gene elements), fits with the fact that drastic decrease in *Klotho* expression by itself has little impact on steady state homeostasis, as no common pathways

are disturbed. We visualized the 20 most significantly deregulated genes in female WT vs. E1 comparison side by side with the other groups as a heatmap (Fig. 4b). In addition to those genes being different between lines, several of them display sexual dimorphism as well (*Jchain*, *Gulo*, and *Hdc*). RNA-seq tracks comparing peak height and exon read distribution (Fig. 4c and Supplementary Fig. 6) reveal an overall trend similar to the qPCR results, but curiously the difference lessens when focused on exon 1 of *Klotho*, except for the low amount of reads in E1 KO females, again indicating presence of a sexually dimorphic promoter element remaining active in E1 KO males (Fig. 4d, e). To ensure the enhancer is not impacting genes further upstream and downstream, we investigated read counts of the four closest neighbors of *Klotho* and did not observe significant differences between experimental groups (Supplementary Table 4). Since *Klotho* deregulation might cause compensatory mechanisms in renal calcium and phosphate handling, we additionally investigated the expression of three ion channel families: *SLC*, *CAC*, and *TRPV* (Supplementary Fig. 7). After filtering out genes of low renal expression (average read count <20), the remaining 339 transporters rarely exceeded 50% increase or decrease in average expression when comparing WT and E1 knockouts within sexes. While presence of some of the highest deregulated genes might still be explained by their relatively low read counts, it is of note that most of the others are highly sexually dimorphic, either at the baseline (Slco1a1, Slc7a13, Slc22a12, and Slc22a28), or through E1 deletion seemingly affecting only one sex (*Slc14a2, Slc25a42,* and *Slc16a14*). Following, we did not find strong

**Fig. 3 | Female E1 knockout mice display significantly lower *Klotho* expression at the baseline and after E1 deletion.** qPCR of renal *Klotho* (**a**) and serum FGF23 levels (**b**) in wildtype and mutant E1 mouse lines. Representative histone modification ChIP-seq of *Klotho* locus histone modifications in female (blue) and male (red) wild type and E1 mice (**c**). Serum phosphate and calcium concentrations in male and female WT and E1 mice (**d**, **e**). Expression is shown as fold change compared to control. Bar = SEM; **a**, **d**, **e** *n* = 4, **b** *n* = 4 for female and *n* = 6 for male mice, Two-way ANOVAs with group mean comparison, **$P < 0.01$, ***$P < 0.001$.

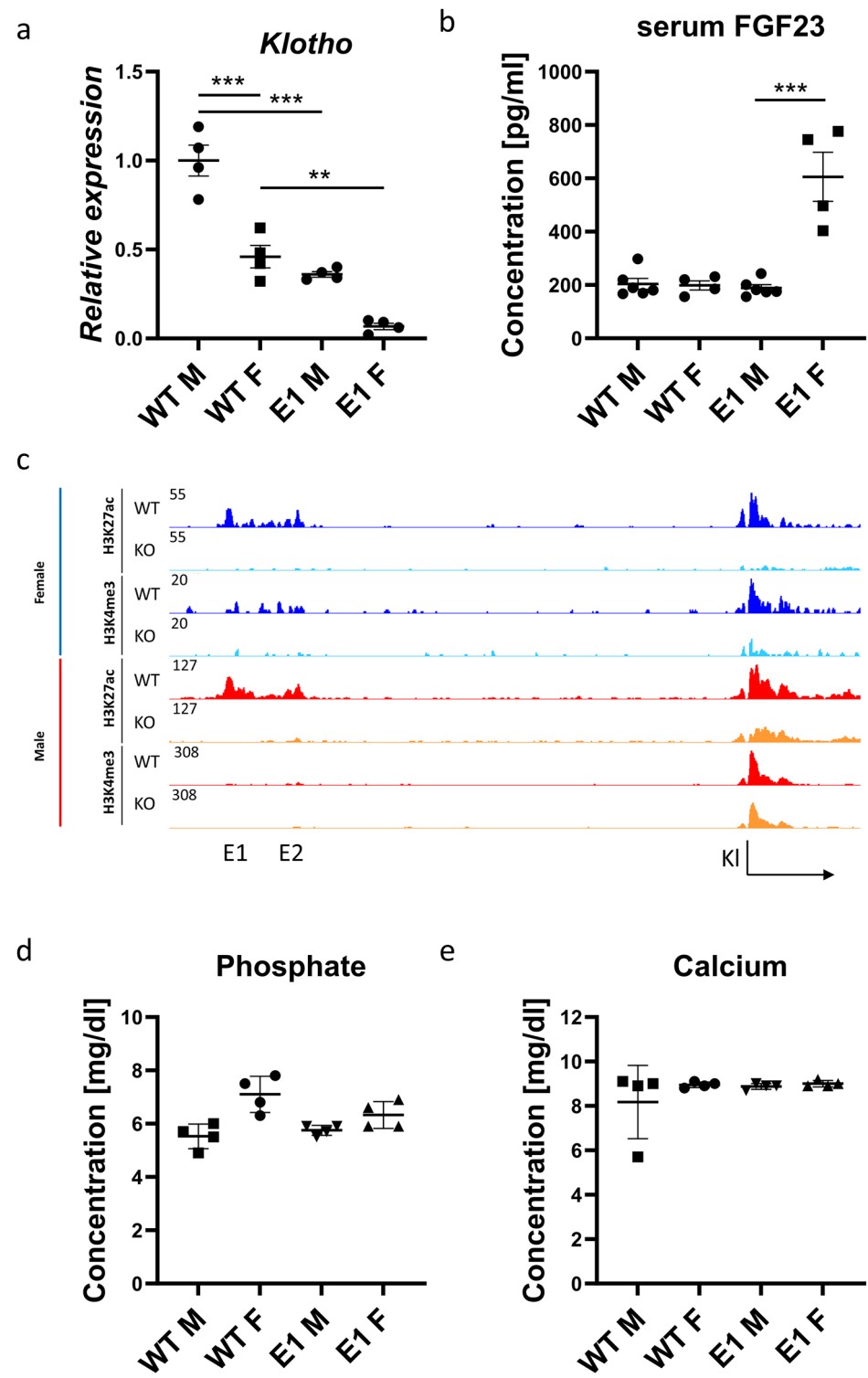

evidence that the serum phosphate and calcium levels in mutant mice are maintained by deregulated renal ion transporters.

**Enhancer deletion decreases renal Klotho protein expression**
To confirm the effects of deletion are not restricted to mRNA and are reflected on protein level, we performed immunohistochemistry on male and female, WT and E1/2 kidneys (Fig. 5 and Supplementary Fig. 8). We observed expected tubular staining of the renal cortex, but none in glomeruli or renal medulla. Staining is visibly stronger in male WT mice compared to WT female. A comparison of the Klotho-positive number of tubules per random microscope photograph additionally confirms staining is more abundant in male mice (Fig. 5b). E1/2 knockout male presents with less staining intensity and tubule number than WT, and we were unable to find positive Klotho staining in female E1/2, resembling qPCR results.

**Impact of acute kidney injury is increased by *Klotho* depletion only in male mice**
Next, recognizing the sexual dimorphism of our model, we performed renal ischemia-reperfusion injury in 3-month-old WT, E1 KO, and E1/

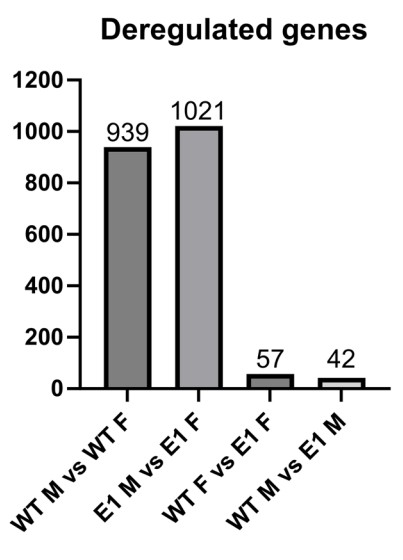

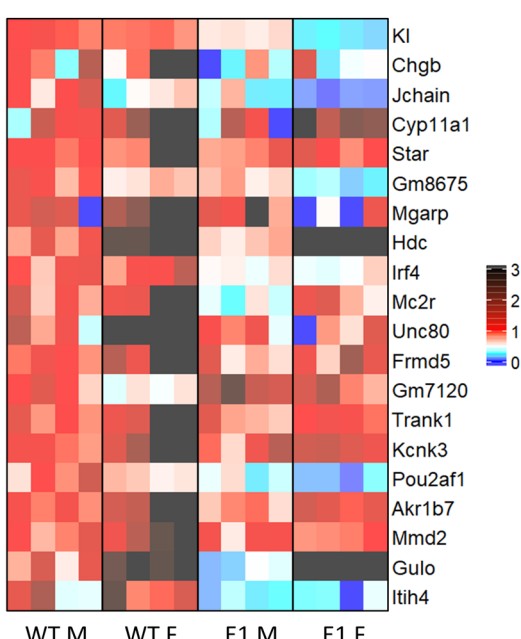

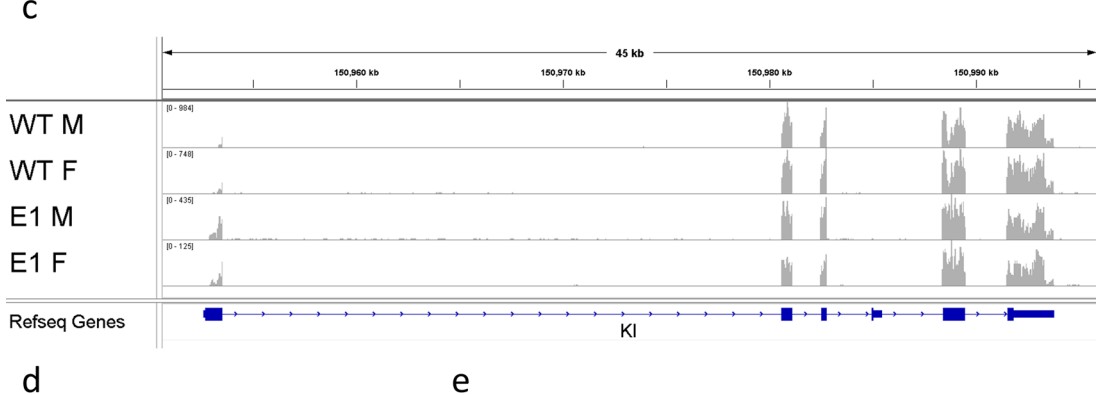

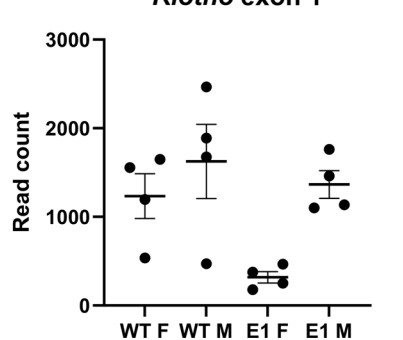

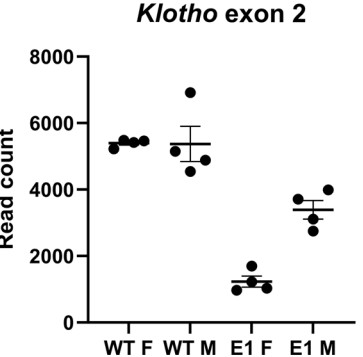

**Fig. 4 | E1 deletion causes a shift in kidney gene expression.** A number of genes were detected by renal RNA-seq analysis was significantly deregulated between WT and E1 KO males and females (**a**). Heatmap of the 20 most deregulated genes in female genotype comparison, normalized to male WT expression (**b**). RNA-seq alignment showing read density at the *Klotho* locus (**c**). Read counts mapping to exons 1 and 2 of *Klotho* in male and female WT and E1 mice (**d**, **e**). Bar = SEM. **d**, **e** *n* = 4.

E2 KO males and females. We chose a relatively severe, 30-min bilateral model to ensure induction of injury in female mice, which are known to be more resilient to AKI[36]. We were able to elicit similar levels of serum creatinine increase and body weight loss, indicating successful

induction of injury (Fig. 6a, b). Mice were euthanized after 24 h and renal *Klotho* and *Havcr1* (Kidney injury molecule 1[37]) expression was measured as direct indicators of renal health and acute injury response. Compared to the baseline of their respective sexes, the decrease of

**Fig. 5 | Renal Klotho protein expression displays sexual dimorphism and is diminished in E1/2 knockout mice.** Representative photographs of renal cortex of male and female WT and E1/2 knockout mice at 200x (left) and 400x (right) total magnification stained for Klotho protein (**a**). Outlined sections are magnified, bar = 50 μm. Quantification and comparison of Klotho-positive tubules in a series of cortical photographs (**b**). Bar = SEM; $n = 10$, non-paired $T$-tests (WT vs E1/2 males, WT male vs female) and unpaired Mann–Whitney tests (WT vs E1/2 females, E1/2 males vs females), ***$P < 0.001$, ****$P < 0.0001$.

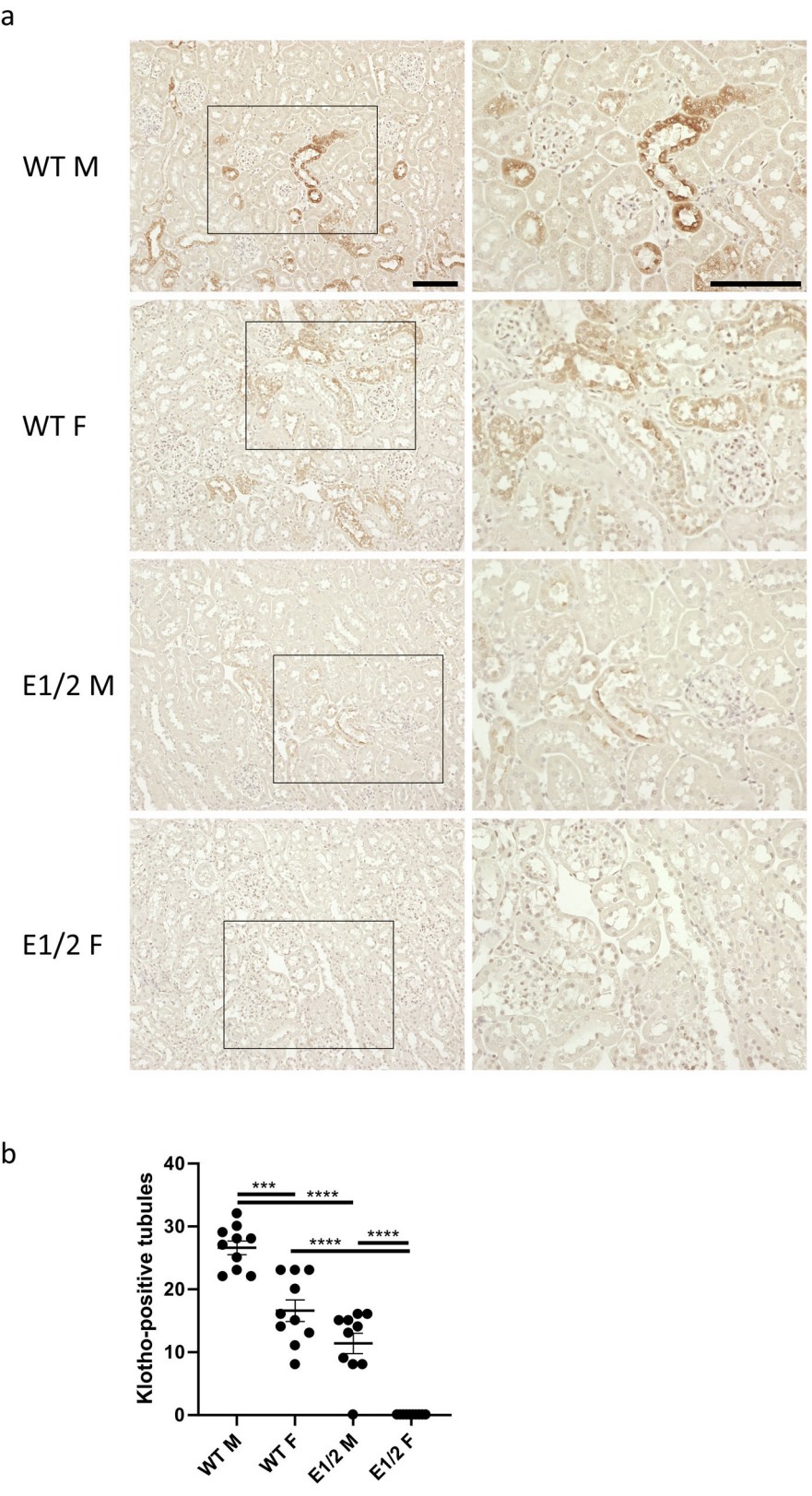

*Klotho* after injury is more pronounced in males than in females, though the final levels after AKI are similar (Fig. 6c, d). In contrast, while we observed an increase in renal *Havcr1* in both sexes, it was relatively much more increased in males, with only mutant males indicating more severe injury than WT controls (Fig. 6e, f).

**_Klotho_ depletion does not exacerbate fibrosis in male or female E1/E2 KO mice**

Finally, we investigated the effects of diminished *Klotho* expression on the development of renal fibrosis. We used 4-month-old male and female WT and E1/E2 KO mice in a 30-min unilateral ischemia-reperfusion model and

**Fig. 6 | Female kidneys remain protected against AKI despite *Klotho* depletion.** Serum creatinine (**a**) and % weight loss (**b**) at 24 h after AKI in male and female wild type, E1 KO and E1/E2 KO mice. Renal *Klotho* (**c, d**) and *Havcr1* (**e, f**) expression 24 h after AKI in male and female wild type, E1 KO and E1/E2 KO mice. Expression is shown as fold change compared to control. Bar = SEM; **a–d** $n = 4$, Two-way ANOVA with group mean comparisons, separate for WT vs. E1 and WT vs. E1/E2, *$P < 0.05$, **$P < 0.01$, ***$P < 0.001$, ****$P < 0.0001$; **e, f** $n = 4$, non-paired *T*-test between IRI WT and IRI E1 groups, *$P < 0.05$.

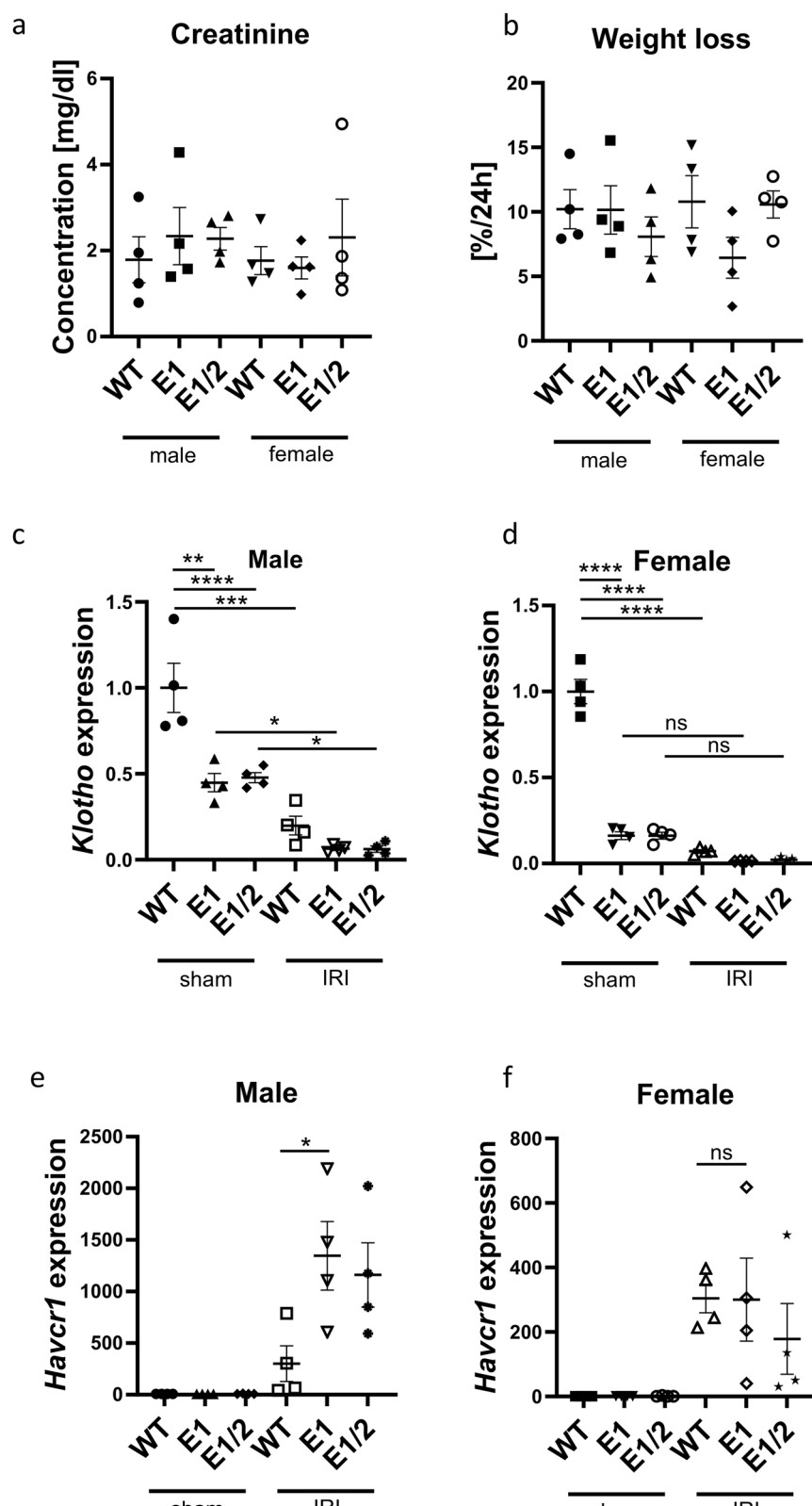

euthanized them after 28 days. Unexpectedly, we saw no further decrease in *Klotho* expression in the injured kidney (Fig. 7a, b). While we saw a marked increase in *Tgfβ* and *Acta2* in male mice, indicating successful induction of fibrosis, no such increase was present in female mice (Fig. 7c–f). The levels of fibrosis markers did not differ between WT and E1/E2 KO mice, and the similar fibrosis development within the same sexes was confirmed by Masson Trichrome staining, with female mice displaying less renal collagen deposition (Fig. 7g, h and Supplementary Fig. 9).

## Discussion

This is the first report to identify and validate the function of a transcription enhancer in the kidney that controls dimorphic *Klotho* expression. We show

**Fig. 7 | *Klotho* depletion alone does not exacerbate fibrosis in double knockout mice.** Renal *Klotho* (**a**, **b**) *Acta2* (**c**, **d**), and *Tgfb* (**e**, **f**) expression 28 days after AKI in injured and contralateral male and female wild type and E1/E2 KO mice. Quantification of Masson Trichrome staining in male and female wild type and E1/E2 KO kidneys (**g**, **h**). Expression is shown as fold change compared to control. Bar = SEM; **a–h** *n* = 6 except male E1/2 group, where *n* = 5 (mouse excluded due to death during surgery recovery, cause undetermined), Two-way ANOVAs with group mean comparisons, **\*\****P* < 0.01, **\*\*\*\****P* < 0.0001.

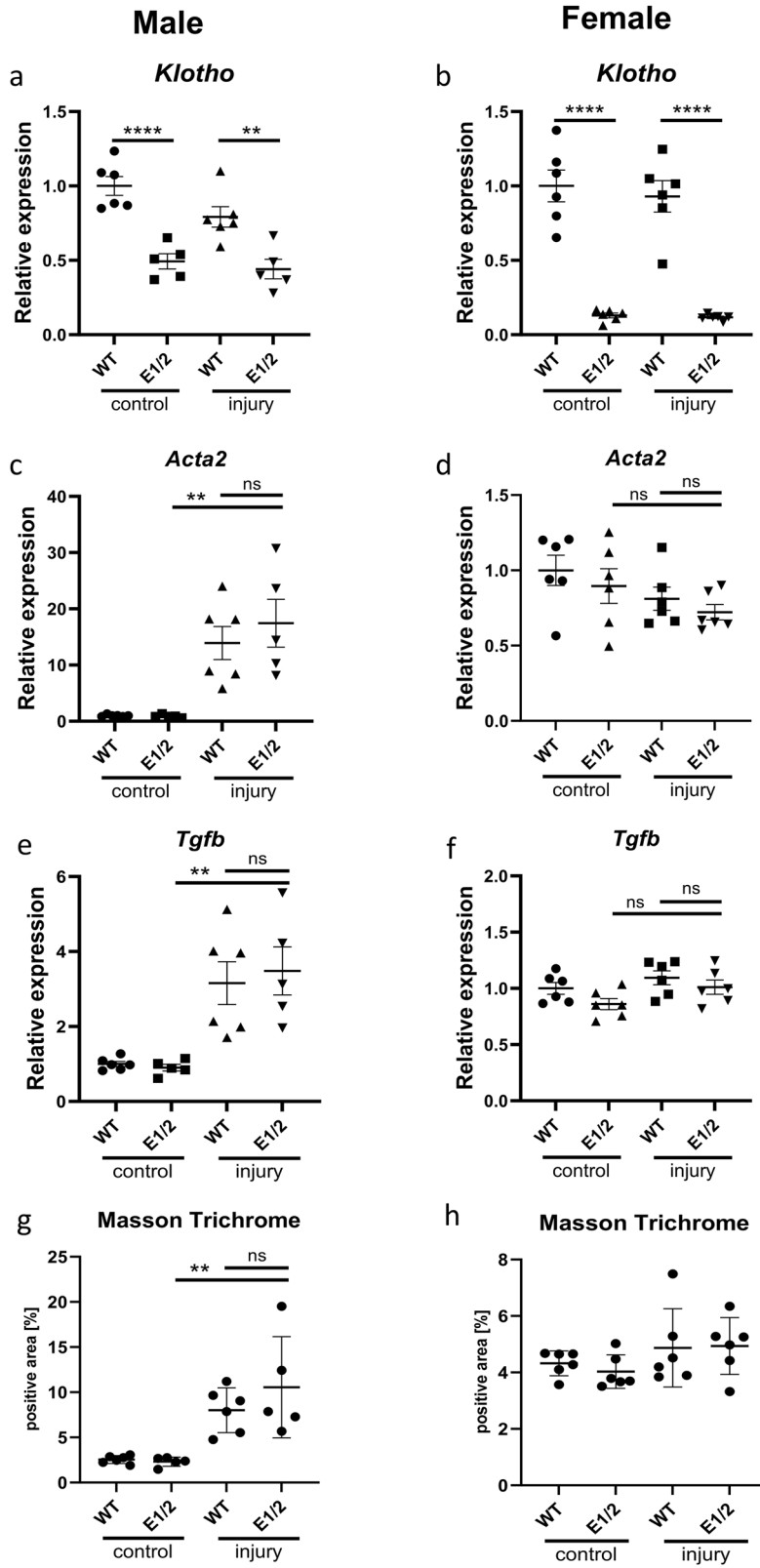

that the principle of excising tissue-specific enhancers allows for a detailed investigation and regulation of target genes. The concept of enhancer disruption and modification contributing to disease development has been well-established in the literature. Possibly the most versatile example of that is enhancer hijacking in cancer, where enhancers are able to activate oncogenes after transposition[38]. Deletions, methylation, or SNP accumulation in enhancers is known to be able to induce a number of congenital and chronic diseases[39–41]. Majority of disease enhancers are first reported as prediction based on ChIP-seq and similar techniques. Although validating the activity of potential enhancers through animal models is essential, it remains infrequent due to its time-consuming nature, expenses, and uncertain outcomes. Literature both reports the potential for significant

impact of the deletions on phenotype in vitro or in vivo[42,43]. and, no less importantly, no differences in knockout strains[44]. To our knowledge, this is the first report on the functionality of an enhancer of a known renal disease marker. The only comparable work we are aware of is the recent detailed dissection of *Cyp27b1* locus enhancers[45], a gene which by itself is not currently widely considered a kidney disease risk factor, but is heavily involved in the PTH-FGF23 axis, just like *Klotho*.

It was well within expectations for deletion of the *Klotho* enhancers to elicit strong dysregulation of transcription, as the activity of those elements decreases after kidney injury in tandem with gene expression[4]. Kidney pathology displays large shifts in its epigenetic landscape, as recently discussed in the literature. For example, as was illustrated for polycystic kidney disease, enhancer activity drives cystogenesis, and the majority of PKD-upregulated genes have associated enhancer elements[5,6]. Enhancers also help maintain renal homeostasis, for example, by establishing the cell identity of podocytes[46]. This process is regulated by the WNT pathway, crucial for kidney health[47]. Because of the sparsity of data, it's hard to pinpoint transcription factors driving renal transcription, and a wide range of contributors like NRF2, FOSL1, or STAT proteins were proposed[48–50]. HNF1B, the key regulator of kidney development, is also intertwined with regulating the aforementioned WNT signaling[51]. HNF1b knockout model is embryonically lethal, while kidney-specific deletion results in premature death due to organ underdevelopment, and heterozygous point mutation mimicking human can result in significant renal deformity[52–54]. We are unaware of any reports indicating direct *Klotho* regulation by HNF1b, as shown here. In fact, while the literature indicates potential HNF1b target genes and discusses its importance, there are only a few examples of validated HNF1b-enhancer interactions. It was reported to directly bind to regulatory elements in *Lef1* and *Axin2* gene loci in vitro[55], and several more targets were indicated in a *Xenopus* model[56]. HNF1b itself is regulated by a Pax-8-responsive enhancer, and thus might be a promising target in a subsequent study[57]. Further, human ChIP-seq data suggests the presence of conserved acetylation peaks upstream of *Klotho*, where HNF1b motifs are present as well, indicating a potentially analogous regulatory mechanism. Identification of HNF1b as a regulator of *Klotho* expression binding directly to DNA will enable a more detailed dissection of the complex mechanism of renal gene expression. Through subsequent ChIP-seq experiments, it might be possible to identify additional transcription factors and cofactors binding indirectly and necessary for this enhancer-promoter interaction. This will help better inform therapeutic strategies by recognizing potential interactions, such as HNF1a sharing molecular machinery with HNF1b and being a known mutation target in cancer[58,59]. It is worth noting that there are several other studies indicating transcription factors regulating *Klotho* expression. Proximal tubule expression of Fosl1 attenuates renal injury in an acute setting[49], Egr-1[60], Sp1[61], and PPAR-gamma[62] can regulate Klotho in vitro, and OCT-1 is a porcine *Klotho* regulator[63]. Other epigenetic mechanisms, like promoter methylation and miRNA activity, have been implicated in *Klotho* regulation as well[64–66].

Investigation of *Klotho* depletion is the less common approach in an injury setting, as since the discovery of its anti-aging properties, a variety of research fields have used Klotho supplementation to reverse disease progression and attenuate injury. Klotho proved to be an effective therapeutic agent in several disease models such as bone regeneration[67], pulmonary fibrosis[68], and myocardial ischemia[69]. The need for this supplementation could be avoided if the mechanisms regulating native *Klotho* expression were better understood. So far, attempts to dissect the epigenetics of the *Klotho* locus and preserve its levels were limited to non-specific treatments, such as HDAC inhibition[70]. *Klotho* expression is usually considered to be kidney-specific. The organ not only boasts the highest expression of the membrane protein, but also is responsible for the vast majority of circulating protein. Low levels of expression are observed in other tissues, such as the parathyroid gland and liver Klotho has been suggested to contribute to rodent longevity[71]. Since Klotho expression in other tissues is negligible and the ChIP-seq datasets to date do not indicate the co-presence of enhancer and promoter marks in extrarenal tissues, it is challenging to

assess to what degree our intervention affected those secondary expression locations.

What we know, is that the deletion of the validated E1 enhancer impacted the kidneys in a sexually dimorphic manner. Descriptions and validation of such dimorphic enhancers in the literature are sparse, as most studies focus on single-sex experimental animals or conduct global analyses only indicating potential loci of interest[72,73]. The effects of E1 enhancer deletion only partially resembled those shown in tissue-specific *Klotho* knockouts. While tissue-specific knockouts lack an exon but are still able to initiate the making of a dysfunctional protein, our model allows the cells to maintain low *Klotho* expression levels. Since the gene promoter can function independently of enhancer, mice lacking the E1 enhancer potentially allow for additional compensatory effects at the promoter, which we saw evidenced more strongly in male mice. Despite a marked decrease in *Klotho* mRNA, normal phenotype remained mostly intact. Both male and female tamoxifen-inducible proximal tubule-specific knockouts reported by Takeshita had significantly lower body mass than controls and had impacted phosphate homeostasis despite the renal Klotho still being present through distal tubule expression[21]. Without distinction between male and female mice, the decrease in expression was approximately 70%. Targeted deletion of the gene in the distal tubule, thought to be the main source of Klotho, resulted in no differences in body weight but still impacted phosphate handling caused by similar to E1 KO 50% decrease in expression level[23]. In the same distal tubule-specific Ksp/Kl$^{-/-}$ mouse strain, soluble Klotho protein level was reported to decrease to nearly zero, which predisposes mice to aortic aneurysms[20]. Regrettably, the lack of comprehensive information on the sexual dimorphism of the other models complicates the identification of factors responsible for the disparities between the mutant strains. There are two main possible explanations for the lack of clear physiological changes in E1 and E1/2 knockout mice. One, similarly to the creators of the Ksp/Kl$^{-/-}$ mouse strain, we can speculate that the depletion of *Klotho* is not sufficient to cause systemic phosphate toxicity[20]. Otherwise, an unknown compensatory effect might be present. It is possible the excess phosphate is excreted with urine, and that one of the deregulated genes performs unknown as of now function, or that the mechanism is dependent on miRNA, known to regulate both *Klotho* and calcium homeostasis[74,75]. A key area of focus of subsequent studies should be establishing the functional role of sexual dimorphism. As evidenced in our results, sex differences in WT renal expression of *Klotho* are better visible in RT-qPCR than in bulk RNA-seq. Finally, we present protein expression data further consolidating the presence of differences in cortical Klotho expression between sexes. We observe less intense and abundant IHC staining in female WT mice, and no visible staining in mutant females.

In addition, to our knowledge, none of those models had been subjected to renal injury models. Higher susceptibility to injury can be assumed based on deregulated phosphate-calcium homeostasis involving hypertension and hyperphosphatemia, but as our results show, *Klotho* deficiency alone might not be enough to disturb phosphate/calcium homeostasis and thus be the defining factor in injury response, especially in females. Our data indicates only a slight predisposition to injury in E1 mutant mice, however considering similar *Havcr1* expression trend in E1 and E1/E2 double knockout mice and similar levels of *Klotho* decrease in those strains, subsequent studies focused on injury modulation should elucidate the extent of this susceptibility. In a chronic disease setting, isolated *Klotho* depletion does not worsen the outcome in mice, suggesting that in a clinical setting, it's an effect, not a cause, of declining health, or that an unknown compensatory effect exists, allowing for the normal repair process.

Due to the female mice being more resilient than males in the AKI setting, they are far less often used in injury models. This protective phenomenon is mirrored in humans, and though historically the female sex was considered to be a risk factor, current analyses indicate high testosterone, rather than the absence of estrogen, to be an important contributor to AKI development[26,76,77]. The human differences in *Klotho* expression remain to be investigated in detail. Recent studies show no significant difference between the levels of soluble Klotho in the serum[78], though high testosterone

has also been associated with high soluble Klotho levels[79]. The disparities might be caused by different assays used to measure serum Klotho. A study comparing two different immunoassays found that either there is no difference between men and women, or that serum Klotho levels are higher in women[80]. To further confound those findings, different primate species, such as bonobos and chimpanzees, can display the opposite direction of sexually dimorphic *Klotho* expression[81]. While the hypothesis was posed that those differences might stem from androgen and estrogen regulation of Klotho, the direct interaction between androgen receptor (AR) and *Klotho* is yet to be confirmed, as the AR motif indicated in *Klotho* promoter is not well conserved, and the only evidence for this interaction comes from rat in vitro experiments[82]. A recent report by Xiong et al. dissects in detail the range of renal transcriptome's sexual dimorphism and its dependence on androgen receptor signaling but observes no significant change in Klotho expression or dimorphism caused by orchiectomy/ovariectomy or nephron-specific androgen and estrogen receptor knockouts in bulk RNA-seq analysis[83].

The information our model adds to the renal injury literature is significant. A Marked decrease in Klotho increased the AKI severity in male mice but is not solely responsible for the degree of injury, as female mice were less affected despite the 90% decrease in renal *Klotho* expression. One of the few human studies available indicates that the renal Klotho levels correspond to the injury severity[84]. There are three possible explanations for our finding: (1) either the soluble and membrane Klotho proteins have distinct predictive powers, (2) it's not the pre- but only post-injury Klotho that allows for differentiating injury levels, (3) the sexual dimorphism in the injury response itself stems from Klotho-independent factors. More surprisingly, we were unable to confirm any effect of Klotho depletion on the development of fibrosis. While we observed increased fibrotic marker expression in our model in male mice, females remained undisturbed by the injury. In both sexes, there was no further decrease in Klotho expression after injury, though there is a possibility of the expression levels recovering during the 4-week fibrosis development period. This, however, would still contradict the large body of literature clearly indicating that decreased Klotho has a strong impact on the progression of chronic kidney disease and renal fibrosis[85,86]. The evidence encompasses Klotho-derived peptides for the treatment of fibrosis and microRNA silencing of *Klotho* worsening the injury[87,88]. One possible explanation is that our model requires a follow-up period longer than a month to be able to differentiate between experimental groups, though the lack of any decrease in *Klotho* expression at the day 28 timepoint makes it less probable. Literature indicates that fibrosis marker expression (*TGFb*, *Col1a1*) peaks 72 h after injury and remains elevated at least 7 days later, but with *Klotho* level remaining at baseline, there is no indication enhancer deletion would worsen the development of fibrosis[89]. Finally, while sex hormones might explain the sexual dimorphism of Klotho expression, they are also likely partially responsible for female protection, helping attenuate the effects of its depletion[90,91]. A number of different mechanisms can furrher contribute to this effect, such as higher level of endithelial disruption in males[92], Sirutin protein family expression[93,94] or mitochondrial health[95]. The degree of those pathways interacting with each other remains a field of active study.

In conclusion, our study highlights the importance of taking sexual dimorphism into consideration while exploring the role of epigenetic modifications. Our findings clearly demonstrate the involvement of Klotho in the FGF23 signaling pathway and likely kidney injury response, though those observations are dependent on the sex of research subjects. We demonstrate the value of validating enhancer function by their selective deletion and show that genetically manipulating regulatory cis-elements can change gene expression levels and result in unpredictable effects on phenotype.

## Methods
### Mice
**General care**. All animals were housed in the same environmentally controlled room (22–24 °C, with 50 ± 5% humidity and 12 h/12 h light–dark cycle) and handled according to the Guide for the Care and Use of Laboratory Animals (8th edition) and all animal experiments were approved by the Animal Care and Use Committee (ACUC) of National Institute of Diabetes and Digestive and Kidney Diseases (NIDDK, MD) and performed under the NIDDK animal protocol K089-LGP-20. All mice were 12–16 weeks old at the time of respective experiments unless stated otherwise.

**Generation of mutant mice**. CRISPR-Cas9 targeted mice were generated using B6D2F1/J (deletion of enhancer 1, E1) or C57BL/6N (E2) (Charles River) by the transgenic core of the National Heart, Lung, and Blood Institute (NHLBI). Single E1 KO mouse strains were developed using one sgRNA, while E2 KO were generated using two sgRNAs. Due to the proximity of the enhancers, E1/E2 double knockouts were obtained by using E2 sgRNAs on homozygous E1 KO mice rather than generating them independently. All mice were genotyped by PCR amplification and Sanger sequencing (Quintara Biosciences) with genomic DNA from mouse tails. Sequences of genotyping primers and sgRNAs are in Supplementary Table 1. Sequences of all deletions are available in Supplementary Table 2. The background of WT mice in each experiment reflects that of the mutant mice to avoid any impact of the founder strain on results.

**Ischemia-reperfusion surgery**. Randomized litters of homozygous-bred mice were used for all the procedures. We chose ~3–4 months old mice to balance avoiding mortality in our severe model (2.3% for procedures presented) while aiming for more pronounced injury than in young animals. To perform warm renal ischemia-reperfusion, in random order and with the surgeon blinded to genotype, but not sex, mice were anesthetized with ketamine/xylazine mix (100 mg/kg and 10 mg/kg respectively). Hair was removed from the mouse retroperitoneal area using sterilized electrical clippers, and skin was cleared and prepared using betadine and ethanol swabs. Next, the mice were placed over a temperature-controlled heating pad maintained at 38 °C. The core temperature of the mice was sustained at ~35.5–36 °C as measured by a rectal probe. Renal Ischemia was induced by clamping the renal artery for 30 min, either uni- or bilaterally. Then, the clamp was removed, and the skin was closed using sterile wound clips, which were removed at the time of euthanasia or 14 days after surgery depending on follow-up time. Finally, the animals were injected with 1 ml saline to replenish fluids, provided analgesia (sustained release Buprenorphine, 1 mg/kg), and allowed to recover in a cage heated to ~37 °C until anesthesia wore off.

**Ovariectomy**. Three-week-old female WT mice were used for the procedure. Mice were prepared for surgery as above. A dorsal skin incision was made parallel and lateral to the spine from the mid-thoracic curvature to the end of curvature, through subcutaneous tissue and muscle to the abdominal cavity. The fat pads were lifted outwards using forceps to exteriorize the ovary. The ovary was then removed by cauterizing between the ovary and oviduct and removing all ovarian tissue. Incision was closed using 6-0 absorbable interrupted suture and staples, then mice were given analgesic and put in recovery as above. Staples were removed 14 days after surgery.

### Chromatin immunoprecipitation and sequencing (ChIP-seq) and data analysis
Renal tissues were frozen with dry ice and stored at -80°C prior to use. Each ChIP library was prepared by combining tissue from four animals of the same genotype to reduce variability. After grinding the tissue with mortar and pestle under liquid nitrogen, chromatin was fixed with formaldehyde (1% final concentration) for 15 min at room temperature and then quenched with glycine (0.125 M final concentration). Nuclei were isolated with Farnham Lysis Buffer (5 mM PIPES pH 8.0, 85 mM KCl, 0.5% NP-40, PMSF, and proteinase inhibitor cocktails). The chromatin was fragmented to 200–500 bp using sonicator 3000 (20 s pulse/30 s rest, 24 min active time, Misonix Sonicators) and further lysed in RIPA buffer. Approximately one milligram of chromatin was immunoprecipitated with Dynabeads Protein

A (Novex) coated with antibodies. The following antibodies were used for ChIP-seq: H3K27ac (Abcam, ab4729), H3K4me3 (Millipore, 07-473), and HNF1b (Invitrogen, 720259). Then, 5–10 ug of antibodies were added to 1 mg of total proteins. After serial bead washes, ChIP DNA was reverse crosslinked at 65 °C overnight in the presence of 1% SDS and 1 mg/ml of Proteinase K (Invitrogen), and DNA was purified with QIAquick PCR Purification Kit (Qiagen). The DNA fragments were blunt-ended and ligated to the Illumina index using the NEBNext Ultra II DNA Library Prep kit for Illumina (New England BioLabs). Libraries for next-generation sequencing were prepared and sequenced with a NovaSeq 6000 instrument (Illumina). The quality of received raw data files was assessed with FastQC and data under quality threshold was removed using Trimmomatic[96]. Bowtie2 was used for alignment using the reference genome mm10 and MACS2 for peak calling[97,98]. Integrative Genomics Viewer was used for data visualization[99]. FIMO[32] and UCSC Genome Browser[33] were used to detect potential HNF1b binding sites in the human sequence upstream of *Klotho* and to visualize sequence conservation between species. In addition to original data, GEO series GSE114292 and GSE104907 were used to obtain H3K27ac, H3K4me3, Pol II, GR, and Esrry data visible in Fig.1a, b and GSE129585 was used for human H3K27ac track in Supplementary Fig. 2. Data quality were reported in Supplementary Table 3.

### Bulk RNA sequencing (total RNA-seq) and data analysis

Bulk RNA was extracted from whole frozen renal tissue from wild-type and mutant mice and purified with RNeasy Plus Mini Kit (Qiagen, 74134). Ribosomal RNA was removed from 1 μg of total RNAs, and cDNA was synthesized using SuperScript III (Invitrogen). Libraries for sequencing were prepared according to the manufacturer's instructions with TruSeq Stranded Total RNA Library Prep Kit with Ribo-Zero Gold (Illumina, RS-122–2301), and paired-end sequencing was done with a NovaSeq 6000 instrument (Illumina). Read quality control was done using FastQC and Trimmomatic[96]. RNA STAR was used to align the reads to mm10 genome[100]. HTSeq and DeSeq2 were used to obtain gene counts and compare genotypes[101,102]. The data were pre-filtered keeping only those genes, which have at least ten reads in total. Genes were categorized as significantly differentially expressed with an adjusted *p* value below 0.05. Differentially expressed genes were visualized with ComplexHeatmap R package[103]. Data quality were reported in Supplementary Table 3.

### RNA isolation and quantitative real-time PCR (qRT-PCR)

Total RNA was extracted from frozen whole or cortical renal tissue of wild-type and mutant mice using a homogenizer and the PureLink RNA Mini kit according to the manufacturer's instructions (Thermo Fisher Scientific). Total RNA (1 μg) was reverse transcribed for 50 min at 50 °C using 50 μM oligo dT and 2 μl of SuperScript III (Thermo Fisher Scientific) in a 20 μl reaction. Quantitative real-time PCR (qRT-PCR) was performed using TaqMan probes: mouse *Klotho* (Mm00502002_m1), *Havcr1* (Mm00506686_m1), *Acta2* (Mm0156133_m1), *Tgfb1* (Mm01178820) *Gapdh* (Mm99999915_g1), Thermo Fisher Scientific on the CFX384 Real-Time PCR Detection System (Bio-Rad) according to the manufacturer's instructions. PCR conditions were 95 °C for 30 s, 95 °C for 15 s, and 60 °C for 30 s for 40 cycles. All reactions were done in duplicate. Relative differences in PCR results were calculated through the CFX Manager software (Bio-Rad) using the comparative cycle threshold (CT) method and normalized to *Gapdh* levels.

### Serum component measurements

Serum FGF23 was measured using ELISA kit according to manufacturer's instructions (CY-4000 Kainos Laboratories) and serum creatinine with a colorimetric kit (Diazyme). Full renal panel was performed by VRL Diagnostics.

### Fibrotic area assessment

Kidneys were fixed in 10% neutral buffered formalin for 24 h, washed, and stored in 70% ethanol. Masson Trichrome staining was performed by

Histoserv. Keyence BZ-9000 microscope was used to take serial, randomized photographs of cortical kidneys at a total of 200x magnification. At least five photographs were taken per animal. Photographs with at least 90% tissue coverage were then processed with countcolors (https://CRAN.R-project.org/package=countcolors) R package to obtain a percentage of photograph occupied by pixels in the blue spectrum reflective of Masson Trichrome stain.

### Klotho immunihistochemistry and quantification

Kidneys were fixed in 10% neutral buffered formalin for 24 h, washed, and stored in 70% ethanol. The preparation of unstained paraffin slides was performed by Histoserv. Slides were rehydrated by washing twice in xylenes, 1:1 xylenes-ethanol solution, twice in 100% ethanol, 95% ethanol, 70% ethanol, 50% ethanol, and twice in water; each washing step lasting 3 min. Next, slides were boiled in Antigen Unmasking Solution (Vector Laboratories) for 10 min and incubated in fresh 3% hydrogen peroxide for 10 min. After washing with water, blocking in 2.5% goat serum (Vector Laboratories) was performed for 1 h at room temperature, after which the sections were incubated in 1:100 Klotho antibody (Cosmo Bio, KO603) in serum, or serum alone (secondary antibody control) overnight at room temperature. Next, slides were washed twice in water, and suspension of the secondary antibody was added (Goat anti-rat IgG, Vector Laboratories, 30032). After an hour of incubation at room temperature, slides were washed, and the DAB substrate (Vector Laboratories) was added for 10 min. After thorough washing with water, hematoxylin solution (Sigma-Aldrich) was added for 3 min, followed by another wash. Slides were dehydrated by following the reverse order of initial washes, ending with xylenes and subsequent mounting with Permount (Fischer Chemical). Keyence BZ-9000 microscope was used to take serial, randomized photographs of cortical kidneys. Ten photographs at 200x magnification were used to quantify Klotho-positive tubules in each kidney.

### Statistics and reproducibility

GraphPad PRISM 10 was used to analyze experimental data. A normal distribution test (Shapiro–Wilk) was performed before assessing the statistical significance of the findings by using appropriate measures detailed in each figure description. Tukey's method was used for multiple comparison correction. All tests used a two-tailed *p* value and statistical significance was set at *p* < 0.05. Levels of statistical significance were described on graphs as follows: *$P < 0.05$, **$P < 0.01$, ***$P < 0.001$, ****$P < 0.0001$. Group sizes are reported in detail in figure descriptions. Error bars on graphs represent the standard error of the mean (SEM). Any statistically analyzed data is derived from biological replicates.

### Reporting summary

Further information on research design is available in the Nature Portfolio Reporting Summary linked to this article.

### Data availability

All ChIP-seq and RNA-seq datasets generated for this study were deposited in Gene Expression Omnibus (GEO) with the accession number GSE243946 (direct link: https://www.ncbi.nlm.nih.gov/geo/query/acc.cgi?acc=GSE243946). In addition to original data, GEO series GSE114292 and GSE104907 were used to obtain H3K27ac, H3K4me3, Pol II, GR, and Esrrγ ChIP-seq data used in Fig. 1a, b. GSE129585 was used for the human H3K27ac track in Supplementary Fig. 2. Further, data necessary to replicate figures, including original photographs, was deposited in Zenodo data sharing repository with the https://doi.org/10.5281/zenodo.10672247[104]. Any additional data or materials are available on request.

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

## Acknowledgements

We thank the NIDDK Genomics Core and the NHLBI Genomics Core for performing NGS, Xiaojie Zhang for performing the ovariectomy surgery, Jeff Reece, and the Advanced Light Microscopy & Image Analysis Core for microscopy advice and equipment. This work utilized the computational resources of the NIH HPC Biowulf cluster (http://hpc.nih.gov). This work was supported by the Intramural Research Programs of National Institute of Diabetes and Digestive and Kidney Diseases, National Institutes of Health, US (J.J., H.K.L., and L.H.), Intramural Research Programs of National Heart, Lung, and Blood Institute, National Institutes of Health, US (C.L.) and the Austrian Science Fund (J.W. - P30373).

## Author contributions

Conceptualization and methodology: J.J., H.K.L., J.W., C.L., and L.H.; Formal analysis and validation, data curation, and visualization: J.J. and H.K.L.; Investigation: J.J.; Resources: C.L. and L.H.; Writing—original draft: J.J.; Writing—review and editing: J.J., H.K.L., J.W., C.L., and L.H.; Supervision, administration, and funding acquisition: J.W. and L.H. All authors approved the final version of the manuscript.

## Funding

## Competing interests

The authors declare no competing interests.
