## [Peer Review File · Communications Biology]

Reviewers' comments:

Reviewer #1 (Remarks to the Author):

Klotho is a critical regulator of mineral metabolism, and its absence causes early aging phenotypes in mice. In the report entitled, “sexually dimorphic renal expression of Klotho is directed by a kidney-specific distal enhancer response to HNF1b,” Jankowski et al. explored cis-regulatory mechanisms of the Klotho gene. The authors identified two putative distal enhancers for its regulation (named E1, and E2, respectively). Very intricately, the authors used a CRISPR/Cas9 targeting technology to generate enhancer knockout mouse lines to determine the functional significance of these two putative enhancers. By comparing the effects of enhancer deletions (ranging from 1744 bp to 311 bp) in E1, they found that the HNF1b binding site in E1 is crucial for directing renal Klotho expression. They further found that E1 is a dominant enhancer element with no obvious functional contribution of E2 for compensating E1 loss or additive impacts on E2 deletion on top of E1 deletion.

Interestingly, the authors surprisingly found no obvious phenotypes in these enhancer knockout mouse lines, even though Kl mRNA expression was significantly suppressed, particularly in female mice. Their data collectively show the critical enhancer function in vivo for directing the Kl gene and the robustness of mineral metabolism in mice. While the paper adds important mechanistic information on Kl gene regulation, this reviewer noted some areas that require improvements to support their conclusion, particularly the sexual dimorphism of Kl gene expression. The profile of this paper would be greatly improved by addressing the following suggestions:

(1) Sexual dimorphic expression of Klotho needs to be well described spatially for mRNA and also at protein levels (tissue and in circulation for soluble Klotho). Single-cell RNA sequencing data of kidneys (Kidney Cell Explorer, Ransick et al., Dev Cell 2019) do not clearly support the sex differences in Klotho gene expression in the kidneys. Indeed, there exists some discrepancy in their data. For example, qPCR shows clear sex differences in Kl expression (Fig. 3a) in WT, but it is not seen in bulk RNAseq analyses (Fig.4d). Can the authors use additional techniques to establish sex differences of Kl in their models? For example, do you see the sex differences in Kl expression in particular cell types? While Kl is expressed along the nephron, it is most highly expressed in distal tubules, particularly distal convoluted tubules, are the effects of sex more evident in these cells or other cell types? This could be explored by in situ hybridization.

(2) The authors discuss the potential mechanisms underlying the milder phenotype compared to distal tubule-selective Kl knockout mice. They suggested that the phenotype could be due to the lack of soluble Klotho protein in distal knockout mice. Then, how about

the enhancer knockouts? To further facilitate the comparisons of reported and present mouse models, can the authors test the protein expression level of Klotho? It would be informative to test both kidney and circulating levels.

(3) In addition to the wild-type animals, these approaches (suggested in #1 and #2) could be extended to investigate if E1 or E2 may differentially affect proximal or distal Kl gene expression and its impacts on protein levels.

(4) The authors did not find obvious changes in mineral metabolism (Ca, P) and body weight loss, while the Kl gene expression was reduced robustly, and the FGF23 level went up high in female E1 knockout. These results could be related to compensation as enhancers were deleted in the germline, and the phenotype may be masked with the compensatory response. Do the authors observe any compensatory increase of renal transporters in the E1/E2 knockout (such as Napt2a, which is upregulated in Pepck-Kl knockout, Trpv4 in Ksp-Kl knockout)? PTH is a sensitive measure for the subtle disturbance of Ca/P homeostasis. The authors could measure serum PTH levels in their models.

(5) I see the Y-axis values for Chip-seq are inconsistent among the conditions (i.e., between sexes in Fig. 3C). Could the authors explain how these Y-axis values were chosen?

(6) Are the two enhancer sequences conserved among species, including humans?

(7) E1 and E2 mice apparently have different backgrounds. How about the choice of control animals? Do you use similar background animals (it seems like the authors are trying to create a control WT cohort from initial heterozygous breeding of E1 knockout)?

Reviewer #2 (Remarks to the Author):

The authors show that the deletion of the distal enhancer decreased Klotho expression in an HNF1-dependent manner. Mice with deletion of the enhancer retain a normal phenotype, although males (not females) develop acute kidney injury. This work is commendable, although several issues remain:

* The sentence "Thanks to next generation sequencing methods, such as ChIP-seq, visualization of enhancer histone modifications and DNA binding proteins, including transcription factors, and RNA Pol II became possible" should be reworded for a more professional tone. Suggested rephrasing: "The advent of next-generation sequencing methods, such as ChIP-seq, has enabled the visualization of enhancer histone modifications, DNA binding proteins, including transcription factors, and RNA Pol II."

* Have the authors examined chromatin accessibility using TCGA data? For instance, the paper on RCC (<https://www.ncbi.nlm.nih.gov/pmc/articles/PMC9867739/>) performs such an analysis. This would be a strong addition to the paper.

- * Why are females more resilient to kidney injury despite lower Klotho levels?
- * The authors should include information on the number of reads obtained and sequencing data quality in the supplementary tables.
- * What statistical test was used for the adjustment of comparisons?
- * Show data regarding the extent of renal injury in younger male versus female animals.
- * For the fibrotic area assessment, what magnification was used for the five areas? These details are important to include.
- * The phrase "We observed that while E2 deletions retain some of the E1 marks, E1 deletion diminishes the E2 peak as well" is vague. Please quantify "some".
- * Do the authors believe that extending the experiment beyond 28 days would have shown a fibrotic phenotype? The elevation of TGFbeta at that timepoint may still be too early to observe fibrosis histologically. They briefly discuss this in the discussion, but further elaboration is needed.
- * Figure 4d: Show read counts of a nearby gene to Klotho as a control to assess how much read depth is contributing.
- * The study could benefit from a more detailed discussion on potential compensatory mechanisms that allow for normal phenotypes despite significant Klotho reduction.
- * The physiological relevance of the findings in human kidney disease is not fully explored and could be addressed in future research.
- * While the critical role of the HNF1b transcription factor in regulating the E1 enhancer was confirmed, the precise interaction between HNF1b and Klotho regulation remains incompletely understood. The partial deletion of the HNF1b binding site significantly affected Klotho expression, while other deletions within the enhancer region did not have the same impact, indicating a complex regulatory mechanism that warrants further investigation. This should be included in the discussion.

Reviewer #3 (Remarks to the Author):

The authors build upon their previous work on transcription enhancers to create novel mouse models of Klotho deficiency by deleting two regions identified as enhancer regions. They then phenotype these knockout mice, assess whether they are more vulnerable because of the lower expression of Klotho in an IRI model, and they find that mice display sexual dimorphism with regard to Klotho expression.

This study, in my view, is relevant and well-executed and the newly generated animal models are potentially valuable research tools. The details in the text would allow for

reproduction of the experiments. I do have some critiques.

Major comments:

- How did the reduction in Klotho gene expression translate to renal Klotho protein expression (Western blot and/or IHC or IF)? This is a relevant question both in the description of the model, and in the IRI experiment. Is there also sexual dimorphism on the protein level?
- Is Klotho expression regulation by HNF1b also relevant in humans? I believe the impact of this study could be increased if an analogous mechanism in humans could be described or even predicted.

Minor comments:

- I would suggest amending the title to clarify that this study, in its current form, is a mouse study on regulation of the murine Klotho gene.
- Please include in the discussion section more relevant literature on what is known about transcriptional regulation of Klotho gene expression in mice and in humans.
- All gene expression data are expressed as relative to a control level. This makes Figure 5c-f and Figure 6a-f more difficult to interpret, as we cannot compare experimental groups properly. I would suggest adjusting the axes as to make gene expression reporting uniform across sex and experimental group.
- Did the authors in their statistical analysis correct for multiple testing?
- Please clarify what is meant by “Investigation of Klotho depletion is a rare approach” in the discussion. Hundreds of studies have phenotyped experimental models of Klotho deficiency in various species and using various techniques.

Detailed Reviewer response

Reviewer #1 (Remarks to the Author):

Klotho is a critical regulator of mineral metabolism, and its absence causes early aging phenotypes in mice. In the report entitled, “sexually dimorphic renal expression of Klotho is directed by a kidney-specific distal enhancer response to HNF1b,” Jankowski et al. explored cis-regulatory mechanisms of the Klotho gene. The authors identified two putative distal enhancers for its regulation (named E1, and E2, respectively). Very intricately, the authors used a CRISPR/Cas9 targeting technology to generate enhancer knockout mouse lines to determine the functional significance of these two putative enhancers. By comparing the effects of enhancer deletions (ranging from 1744 bp to 311 bp) in E1, they found that the HNF1b binding site in E1 is crucial for directing renal Klotho expression. They further found that E1 is a dominant enhancer element with no obvious functional contribution of E2 for compensating E1 loss or additive impacts on E2 deletion on top of E1 deletion.

Interestingly, the authors surprisingly found no obvious phenotypes in these enhancer knockout mouse lines, even though Kl mRNA expression was significantly suppressed, particularly in female mice. Their data collectively show the critical enhancer function in vivo for directing the Kl gene and the robustness of mineral metabolism in mice. While the paper adds important mechanistic information on Kl gene regulation, this Reviewer noted some areas that require improvements to support their conclusion, particularly the sexual dimorphism of Kl gene expression. The profile of this paper would be greatly improved by addressing the following suggestions:

(1) Sexual dimorphic expression of Klotho needs to be well described spatially for mRNA and also at protein levels (tissue and in circulation for soluble Klotho). Single-cell RNA sequencing data of kidneys (Kidney Cell Explorer, Ransick et al., Dev Cell 2019) do not clearly support the sex differences in Klotho gene expression in the kidneys. Indeed, there exists some discrepancy in their data. For example, qPCR shows clear sex differences in Kl expression (Fig. 3a) in WT, but it is not seen in bulk RNAseq analyses (Fig.4d). Can the authors use additional techniques to establish sex differences of Kl in their models? For example, do you see the sex differences in Kl expression in particular cell types? While Kl is expressed along the nephron, it is most highly expressed in distal tubules, particularly distal convoluted tubules, are the effects of sex more evident in these cells or other cell types? This could be explored by in situ hybridization.

Response

We thank the Reviewer for this suggestion. The degree of sexually dimorphic expression of Klotho is indeed still inconclusive in the literature. This is because membrane, soluble and secreted Klotho might be regulated through different mechanisms in different experimental settings, and assays measuring them are not consistent, as we describe in the discussion section (lines 457-462). While single-cell approach might indeed elucidate the importance of membrane Klotho, it might not achieve appropriate sequencing depth to sufficiently dissect gene expression across the various renal tubule segments. Sequencing depth is a notoriously critical issue. That being said, the study provided by

*the Reviewer is available as an interactive database under <https://cello.shinyapps.io/kidneycellexplorer/>, and also indicates that *Klotho* displays sexual dimorphism in proximal tubule, one of the most abundant renal cell types, though it might not reach statistical significance in global analyses. We addressed this in the discussion (lines 437-441): “A key area of focus of subsequent studies should be establishing the functional role of the sexual dimorphism. As evidenced in our results, sex differences in WT renal expression of *Klotho* are more obvious in RT-qPCR than bulk RNA-seq. While the exact reason for it is unknown, published single-cell RNA-seq data indicates sex differences in S2 and S3 proximal tubule expression as well.”*

Finally, RT-qPCR and RNA-seq results might not always align, as one relies on detection of a specific exon junction, and the other detects reads aligned over the whole mRNA length. Our time and financial constraints (30% reduction of NIDDK lab budgets) do not allow development and optimization of an in-situ hybridization protocol. However, we provide a new figure (Fig. 5) demonstrating sexual dimorphism on the protein level; please see questions below for more detailed description.

(2) The authors discuss the potential mechanisms underlying the milder phenotype compared to distal tubule-selective KI knockout mice. They suggested that the phenotype could be due to the lack of soluble *Klotho* protein in distal knockout mice. Then, how about the enhancer knockouts? To further facilitate the comparisons of reported and present mouse models, can the authors test the protein expression level of *Klotho*? It would be informative to test both kidney and circulating levels.

Response

*We thank the Reviewer for this important suggestion. To visualize renal *Klotho*, we performed IHC in mutant and WT, male and female kidneys (Fig. 5, Supplementary Fig. 8). We saw a pattern of cortical epithelial expression, which was less intense in WT female mice compared to WT males. This indicates that sexual dimorphism of membrane *klotho* is present in the kidney and that the difference stems not only from a distinct tubule segment but also from overall change in expression. We quantified *klotho*-positive tubules per random microscope field and saw a decrease in females. We describe those findings in the Results section (lines 327-334):*

“Enhancer deletion decreases renal *Klotho* protein expression

*To confirm the effects of deletion are not restricted to mRNA and are reflected on protein level, we performed immunohistochemistry on male and female, WT and E1/2 kidneys (Fig. 5, Supplementary Fig. 8). We observed expected tubular staining of the renal cortex, but none in glomeruli or renal medulla. Staining is visibly stronger in male WT mice compared to WT female. Comparison of *Klotho*-positive number of tubules per random microscope photograph additionally confirms staining is more abundant in male. E1/2 knockout male presents with less staining intensity and tubule number than WT, and we were unable to find positive *Klotho* staining in female E1/2, resembling qPCR results.”*

*We also refer to this figure in the Discussion (lines 441-443): “Finally, we present protein expression data further consolidating presence of differences in cortical *Klotho**

expression between sexes. We observe less intense and abundant IHC staining in female WT mice, and no visible staining in mutant females.”

Unfortunately, due to time and budget constraints, we are unable to perform a serum Klotho measurement at this time, though we expect our observation would extend to the shorter Klotho isoforms

(3) In addition to the wild-type animals, these approaches (suggested in #1 and #2) could be extended to investigate if E1 or E2 may differentially affect proximal or distal Kl gene expression and its impacts on protein levels.”

Response

We agree with the Reviewer, and we extended the IHC analysis to mutant animals (Fig. 5, Supplementary Fig. 8). Double knockout mice display significantly less tubular Klotho staining than the WT animals. In addition, we were unable to find positive staining in female mutants, overall reflecting our qPCR results. We describe those findings in the Results section as follows (lines 327-334):

“Enhancer deletion decreases renal Klotho protein expression

To confirm the effects of deletion are not restricted to mRNA and are reflected on protein level, we performed immunohistochemistry on male and female, WT and E1/2 kidneys (Fig. 5, Supplementary Fig. 8). We observed expected tubular staining of the renal cortex, but none in glomeruli or renal medulla. Staining is visibly stronger in male WT mice compared to WT female. Comparison of Klotho-positive number of tubules per random microscope photograph additionally confirms staining is more abundant in male. E1/2 knockout male presents with less staining intensity and tubule number than WT, and we were unable to find positive Klotho staining in female E1/2, resembling qPCR results.”

We also refer to this figure in the Discussion (lines 441-443): “Finally, we present protein expression data further consolidating presence of differences in cortical Klotho expression between sexes. We observe less intense and abundant IHC staining in female WT mice, and its complete lack in mutant females.”

(4) The authors did not find obvious changes in mineral metabolism (Ca, P) and body weight loss, while the Kl gene expression was reduced robustly, and the FGF23 level went up high in female E1 knockout. These results could be related to compensation as enhancers were deleted in the germline, and the phenotype may be masked with the compensatory response. Do the authors observe any compensatory increase of renal transporters in the E1/E2 knockout (such as Napt2a, which is upregulated in Pepck-Kl knockout, Trpv4 in Ksp-Kl knockout)? PTH is a sensitive measure for the subtle disturbance of Ca/P homeostasis. The authors could measure serum PTH levels in their models.

Response

While we observed only a few statistically significantly altered genes in our RNA-seq WT vs E1 comparison, we agree that looking at a broader ion channel landscape could provide new information. To do it, we visualized fold change in expression, separately for

male and female mice, of SLC, TRPV and CAC gene families (Supplementary Figure 7) and described those findings in the Results section (lines 317-326): “Since *Klotho* deregulation might cause compensatory mechanisms in renal calcium and phosphate handling, we additionally investigated expression of three ion channel families: SLC, CAC and TRPV (Supplementary Figure 7). After filtering out genes of low renal expression (average read count <20), the remaining 339 transporters rarely exceeded 50% increase or decrease in average expression when comparing WT and E1 knockouts within sexes. While presence of some of the highest deregulated genes might still be explained by their relatively low read counts, it is of note that most of the others are highly sexually dimorphic, either at the baseline (*Slco1a1*, *Slc7a13*, *Slc22a12*, *Slc22a28*), or through E1 deletion seemingly affecting only one sex (*Slc14a2*, *Slc25a42*, *Slc16a14*). Following, we did not find strong evidence that the serum phosphate and calcium levels in mutant mice are maintained by deregulated renal ion transporters.” Most of the perceived fold changes might be caused by to low read counts of those genes, though it is interesting that a significant number of renal transporters display sexual dimorphism themselves. We thank the Reviewer for allowing us to provide this insight to the reader.

(5) I see the Y-axis values for Chip-seq are inconsistent among the conditions (i.e., between sexes in Fig. 3C). Could the authors explain how these Y-axis values were chosen?

Response

We would like to clarify our rationale behind ChIP-seq presentation. In Figure 1, we visualize successful deletion of *Klotho* enhancers thus the Y-values were left at default height to ensure the peaks or lack thereof can be appreciated. In Figure 3, we aim to compare WT and E1 ChIP-seq tracks in pairs, allowing for more detailed distinction between males and females. We believe such presentation allows the reader to easier follow our conclusions without distorting or masking any data.

(6) Are the two enhancer sequences conserved among species, including humans?

Response

We thank the Reviewer for this suggestion and agree that translational value of our study would be enhanced if the enhancers were also present in human data. To answer this questions, we obtained human kidney ChIP-seq data from GEO (GSE129585) and visualized it in Supplementary Fig. 2. While human data lack the direct equivalent of E1 and E2 in the same position relative to *Klotho* promoter, the sequence itself shows a high degree of conservation. Additionally, two H3K27ac peaks are visible further upstream, and as indicated by motif analysis they possess HNF1b binding sites, indicating presence of a potential *Klotho* regulatory mechanism.

We describe those findings in the Results section (lines 260-266): “Next, we visualized the mouse and human *Klotho* acetylation marks to investigate whether we can observe analogous peaks and confirm the role of HNF1b (Supplementary Fig. 2a). We did not observe corresponding peaks, however, two peaks relatively further upstream than E1

were present in human ChIP-seq data. We used FIMO tool⁴⁰ to investigate whether HNF1b motifs are present in those loci and found seven matches, indicating a potential for HNF1b activity. Further, we aligned UCSC genome browser⁴¹ to the same region and saw that both the mouse enhancers and human peaks are relatively conserved among rodents and primates (Supplementary Fig. 2b)."

We refer to those findings in discussion (lines 388-390): "Further, human ChIP-seq data suggests presence of a conserved acetylation peaks upstream of Klotho, where HNF1b motifs are present as well, indicating potentially analogous regulatory mechanism."

(7) E1 and E2 mice apparently have different backgrounds. How about the choice of control animals? Do you use similar background animals (it seems like the authors are trying to create a control WT cohort from initial heterozygous breeding of E1 knockout)?

Response

We appreciate the Reviewer's attention to appropriate controls. We would like to confirm that any experimental groups were compared with appropriate WT mice, meaning C57BL/6N for E2 mice and B6D2F1/J, backcrossed once with C57BL/6N, for E1 and E1/2 knockouts. We added additional clarification in the 'Generation of mutant mice' section of the Methods (lines 112-113): "Background of WT mice in each experiment reflects that of the mutant mice to avoid any impact of the founder strain on results."

Reviewer #2 (Remarks to the Author):

The authors show that the deletion of the distal enhancer decreased *Klotho* expression in an HNF1-dependent manner. Mice with deletion of the enhancer retain a normal phenotype, although males (not females) develop acute kidney injury. This work is commendable, although several issues remain:

* The sentence "Thanks to next generation sequencing methods, such as ChIP-seq, visualization of enhancer histone modifications and DNA binding proteins, including transcription factors, and RNA Pol II became possible" should be reworded for a more professional tone. Suggested rephrasing: "The advent of next-generation sequencing methods, such as ChIP-seq, has enabled the visualization of enhancer histone modifications, DNA binding proteins, including transcription factors, and RNA Pol II."

Response

We thank the Reviewer for this suggestion and implemented it directly in the manuscript.

* Have the authors examined chromatin accessibility using TCGA data? For instance, the paper on RCC () performs such an analysis. This would be a strong addition to the paper.

Response

*We thank the reviewer for the suggestion. We investigated the dataset provided by the Reviewer and found that *Klotho* expression in the RCC tissue does not reflect high levels observed in healthy renal tissue. Similarly, expression of *SLC12a3* and *SLC34a1*, distal and proximal tubule markers, is absent in the RNA-seq analysis provided, suggesting the cancer specimens developed from other tubule segments or underwent enough transformation to prove difficult to be useful in our analysis.*

* Why are females more resilient to kidney injury despite lower *Klotho* levels?

Response

*There is a number of potential explanations for this effect. We expanded the discussion to add following statement (lines 491-495): "Finally, while sex hormones might explain sexual dimorphism of *Klotho* expression, they are also likely partially responsible for female protection, helping attenuate effects of its depletion.^{87,88} A number of different mechanisms can further contribute to this effect, such as higher level of endothelial disruption in males,⁸⁹ Sirutin protein family expression^{90,91} or mitochondrial health.⁹² The degree of those pathways interacting with each other remains a field of active study."*

* The authors should include information on the number of reads obtained and sequencing data quality in the supplementary tables.

Response

We agree with the Reviewer that ensuring high quality of data is crucial. We included a new Supplementary Table 3 to detail: read number, average quality and number of reads

of insufficient quality as measured by the FASTQC tool. All reports are available online at the Zenodo repository (DOI: 10.5281/zenodo.12726086).

* What statistical test was used for the adjustment of comparisons?

Response

Tukey's adjustment, recommended by the statistical analysis software, was used for multiple comparison correction. Appropriate note was added to the Methods section (line 213): "Tukey's method was used for multiple comparison correction."

* Show data regarding the extent of renal injury in younger male versus female animals.

Response

We thank the Reviewer for the suggestion, however the impact of the age on injury response is outside of the scope of this manuscript and we'd like to respectfully suggest that the outcome of this experiment would not justify time, effort and limited resources spent. We do not expect younger animals to display significantly different phenotypes, to the contrary, young rodents are more protected against renal injury models such as IRI (Xu X et al., Aging aggravates long-term renal ischemia-reperfusion injury in a rat model. J Surg Res, 2014, Liu et al., Youthful systemic milieu alleviates renal ischemia-reperfusion injury in elderly mice, Kidney Int, 2018). We purposefully chose to include animals reflecting older age instead of the equivalent of young adult humans to better reflect populations suffering from AKI and CKD in clinical setting and increase translational value of the study.

* For the fibrotic area assessment, what magnification was used for the five areas? These details are important to include.

Response

We now included magnification details in the Methods section and appropriate figure descriptions (lines 189, 208, 820-825). We used total 200x total magnification for quantification purposes, as well as 400x to visualize the details of Klotho staining.

* The phrase "We observed that while E2 deletions retain some of the E1 marks, E1 deletion diminishes the E2 peak as well" is vague. Please quantify "some".

Response

We thank the reviewer for pointing out the vague statement. We reworded it to more direct (lines 245-246): "We observed that while E2 deletion retains E1 marks, E1 deletion visibly diminishes E2 peak as well (Fig. 1d)."

* Do the authors believe that extending the experiment beyond 28 days would have shown a fibrotic phenotype? The elevation of TGFbeta at that timepoint may still be too early to observe fibrosis histologically. They briefly discuss this in the discussion, but further elaboration is needed.

Response

While we do not expect divergence of fibrosis development in longer follow-up study after the same unilateral injury, we agree that it is important to provide further context. We added the following to the Discussion section (lines 488-491): “Literature indicates that fibrosis marker expression (TGFb, Col1a1) peaks 72 hours after injury and remains elevated at least seven days later, but with Klotho level remaining at baseline, there is no indication enhancer deletion would worsen development of fibrosis.”⁸⁷

* Figure 4d: Show read counts of a nearby gene to Klotho as a control to assess how much read depth is contributing.

Response

We thank the reviewer for this suggestion and added Supplementary Table 4, describing read counts of genes flanking Klotho. It is now referred to in the Results section (lines 314-317): “To ensure the enhancer is not impacting genes further upstream and downstream, we investigated read counts of the four closest neighbors of Klotho and did not observe significant differences between experimental groups (Supplementary Table S4).”

* The study could benefit from a more detailed discussion on potential compensatory mechanisms that allow for normal phenotypes despite significant Klotho reduction.

Response

We extended the Discussion section to provide additional information (lines 431-437): “There are two main possible explanation for lack of clear physiological changes in E1 and E1/2 knockout mice. One, similarly to the creators of the Ksp/Kl-/- mouse strain, we can speculate that the depletion of Klotho is not sufficient to cause systemic phosphate toxicity.²⁰ Otherwise, an unknown compensatory effect might be present. It is possible the excess phosphate is excreted with urine, and that one of the deregulated genes performs unknown as of now function, or that the mechanism is dependent on miRNA, known to regulate both Klotho and calcium homeostasis.”^{73,74}

* The physiological relevance of the findings in human kidney disease is not fully explored and could be addressed in future research.

Response

We agree with the Reviewer that future research should focus on translational value of our research, and we hope that this manuscript will lay foundation for human research.

* While the critical role of the HNF1b transcription factor in regulating the E1 enhancer was confirmed, the precise interaction between HNF1b and Klotho regulation remains incompletely understood. The partial deletion of the HNF1b binding site significantly affected Klotho expression, while other deletions within the enhancer region did not have the same impact, indicating a complex regulatory mechanism that warrants further investigation. This should be included in the discussion.

Response

We agree with the Reviewer, and we believe we have already strongly stated that HNF1b is at the foundation of Klotho regulation by its enhancer. To reinforce this fact, we added the following to the Discussion section (lines 388-395): “Further, human ChIP-seq data suggests presence of a conserved acetylation peaks upstream of Klotho, where HNF1b motifs are present as well, indicating potentially analogous regulatory mechanism. Identification of HNF1b as a regulator of Klotho expression binding directly to DNA will enable more detailed dissection of the complex mechanism of renal gene expression. Through subsequent ChIP-seq experiments, it might be possible to identify additional transcription factors and cofactors binding indirectly and necessary for this enhancer-promoter interaction. This will help better inform therapeutic strategies by recognizing potential interactions, such as HNF1a sharing molecular machinery with HNF1b and being a known mutation target in cancer.^{66,67”}

Reviewer #3 (Remarks to the Author):

The authors build upon their previous work on transcription enhancers to create novel mouse models of Klotho deficiency by deleting two regions identified as enhancer regions. They then phenotype these knockout mice, assess whether they are more vulnerable because of the lower expression of Klotho in an IRI model, and they find that mice display sexual dimorphism with regard to Klotho expression.

This study, in my view, is relevant and well-executed and the newly generated animal models are potentially valuable research tools. The details in the text would allow for reproduction of the experiments. I do have some critiques.

Major comments:

- How did the reduction in Klotho gene expression translate to renal Klotho protein expression (Western blot and/or IHC or IF)? This is a relevant question both in the description of the model, and in the IRI experiment. Is there also sexual dimorphism on the protein level?

Response

We acknowledge this is a crucial question for our study. To visualize renal Klotho, we performed IHC in mutant and WT, male and female kidneys (Fig. 5, Supplementary Fig. 8). We saw a pattern of cortical epithelial expression, which was less intense in WT female mice compared to WT males. This indicates that sexual dimorphism of membrane klotho is present in the kidney and that the difference stems not only from a distinct tubule segment but also from overall change in expression. We quantified klotho-positive tubules per random microscope field and saw a decrease in females. We describe those findings in the Results section (lines 327-334):

“Enhancer deletion decreases renal Klotho protein expression

To confirm the effects of deletion are not restricted to mRNA and are reflected on protein level, we performed immunohistochemistry on male and female, WT and E1/2 kidneys (Fig. 5, Supplementary Fig. 8). We observed expected tubular staining of the renal cortex, but none in glomeruli or renal medulla. Staining is visibly stronger in male WT mice compared to WT female. Comparison of Klotho-positive number of tubules per random microscope photograph additionally confirms staining is more abundant in male. E1/2 knockout male presents with less staining intensity and tubule number than WT, and we were unable to find positive Klotho staining in female E1/2, resembling qPCR results.

We also refer to this figure in the Discussion (lines 441-443): “Finally, we present protein expression data further consolidating differences in cortical Klotho expression between sexes.”

- Is Klotho expression regulation by HNF1b also relevant in humans? I believe the impact of this study could be increased if an analogous mechanism in humans could be described or even predicted.

Response

We thank the Reviewer for this suggestion and agree that translational value of our study would be enhanced if the enhancers were also present in human data and if HNF1b could be indicated as a regulatory factor. To answer this questions, we obtained human kidney ChIP-seq data from GEO (GSE129585) and visualized it in Supplementary Fig. 2. While human data lack the direct equivalent of E1 and E2 in the same position relative to Klotho promoter, the sequence itself shows a high degree of conservation. Additionally, two H3K27ac peaks are visible further upstream, and as indicated by motif analysis they possess HNF1b binding sites, indicating presence of a potential Klotho regulatory mechanism.

We describe those findings in the Results section (lines 260-266): “Next, we visualized the mouse and human Klotho acetylation marks to investigate whether we can observe analogous peaks and confirm the role of HNF1b (Supplementary Fig. 2a). We did not observe corresponding peaks, however, two peaks relatively further upstream than E1 were present in human ChIP-seq data. We used FIMO tool⁴⁰ to investigate whether HNF1b motifs are present in those loci and found seven matches, indicating a potential for HNF1b activity. Further, we aligned UCSC genome browser⁴¹ to the same region and saw that both the mouse enhancers and human peaks are relatively conserved among rodents and primates (Supplementary Fig. 2b).”

We refer to those findings in discussion (lines 388-390): “Further, human ChIP-seq data suggests presence of a conserved acetylation peaks upstream of Klotho, where HNF1b motifs are present as well, indicating potentially analogous regulatory mechanism.”

Minor comments:

- I would suggest amending the title to clarify that this study, in its current form, is a mouse study on regulation of the murine Klotho gene.

Response

*We agree that a more specific title would be helpful for accurate search and indexing purposes and changed the title to: “Sexually dimorphic renal expression of **mouse** Klotho is directed by a kidney-specific distal enhancer responsive to HNF1b”.*

- Please include in the discussion section more relevant literature on what is known about transcriptional regulation of Klotho gene expression in mice and in humans.

Response

We agree that expanding on Klotho regulation by other transcriptional mechanisms would provide broader context to our finding. We included following in the discussion section

(lines 395-399): *“It is worth noting that there are several other studies indicating transcription factors regulating Klotho expression. Proximal tubule expression of Fos11 attenuates renal injury in an acute setting⁵⁷, Egr-1⁶⁸, Sp1⁶⁹ and PPAR-gamma⁷⁰ can regulate Klotho in vitro, and OCT-1 is a porcine Klotho regulator.⁷¹ Other epigenetic mechanisms, like promoter methylation and miRNA activity have been implicated in Klotho regulation as well.⁷²⁻⁷⁴”*

- All gene expression data are expressed as relative to a control level. This makes Figure 5c-f and Figure 6a-f more difficult to interpret, as we cannot compare experimental groups properly. I would suggest adjusting the axes as to make gene expression reporting uniform across sex and experimental group.

Response

We appreciate the Reviewer’s comment but would like to respectfully suggest that adjusting Y axes to data represented on the particular graph allows for better visualization than making them uniform across experimental conditions.

- Did the authors in their statistical analysis correct for multiple testing?

Response

Tukey’s adjustment, recommended by the statistical analysis software, was used for multiple comparison correction. Appropriate note was added to the Methods section (line 213): “Tukey’s method was used for multiple comparison correction.”

- Please clarify what is meant by “Investigation of Klotho depletion is a rare approach” in the discussion. Hundreds of studies have phenotyped experimental models of Klotho deficiency in various species and using various techniques.

Response

*We thank the Reviewer for the opportunity to clarify. Our intention was to indicate that in an injury setting, studies tend to focus on supplementation of Klotho and observing its therapeutic effect rather than investigating effects of its deficiency. The wording was changed to (line 400): “Investigation of Klotho depletion is **the less common approach in an injury setting**”.*

REVIEWERS' COMMENTS:

Reviewer #1 (Remarks to the Author):

The authors responded well to most of my suggestions. I only have one minor suggestion to the authors.

Authors' response #1, "That being said, the study provided by the Reviewer is available as an interactive database under <https://cello.shinyapps.io/kidneycellexplorer/> and also indicates that Klotho displays sexual dimorphism in the proximal tubule, one of the most abundant renal cell types, though it might not reach statistical significance in global analyses."

Reviewer's comment: I acknowledge the authors' efforts to perform additional immunostaining analyses for sex differences of Klotho protein (Figure 5). Based on this beautiful staining data, I agree that Kl protein expression is sexually dimorphic, and it is apparently regulated by the enhancer they identified. These are very interesting findings and reinforce their conclusion. However, I suggest removing the following sentence (lines 440-441, "While the exact reason for it is unknown, published single-cell RNA-seq data indicates sex differences in S2 and S3 proximal tubule expression as well."). This suggestion is based on the fact that the dataset actually does not show the sex differences in Kl gene expression in proximal tubule cells. See Table S2 of the paper (PMID: 31689386; Female PT, cluster #0,3,9. No Kl in Top 400-500 genes).

Reviewer #2 (Remarks to the Author):

The authors have addressed most of my comments.

Reviewer #3 (Remarks to the Author):

In my opinion, in performing additional experiments and analyses, the authors have greatly improved their manuscript, which describes a valuable model and has important implications with regard to the regulation of Klotho gene expression.

Overall, I have no major comments on the manuscript.